# Repurposing Type I-A CRISPR-Cas3 for a robust diagnosis of human papillomavirus (HPV)
Tao Hu[1], Quanquan Ji[2], Xinxin Ke[1], Hufeng Zhou [3], Senfeng Zhang[4], Shengsheng Ma[4], Chenlin Yu[5], Wenjun Ju[5], Meiling Lu[5], Yu Lin[6], Yangjing Ou[6], Yingsi Zhou [7] ✉, Yibei Xiao [5] ✉, Chunlong Xu [8,9,10] ✉ & Chunyi Hu [4,11,12] ✉

R-loop-triggered collateral single-stranded DNA (ssDNA) nuclease activity within Class 1 Type I CRISPR-Cas systems holds immense potential for nucleic acid detection. However, the hyperactive ssDNase activity of Cas3 introduces unwanted noise and false-positive results. In this study, we identified a novel Type I-A Cas3 variant derived from *Thermococcus siculi*, which remains in an auto-inhibited state until it is triggered by Cascade complex and R-loop formation. This Type I-A CRISPR-Cas3 system not only exhibits an expanded protospacer adjacent motif (PAM) recognition capability but also demonstrates remarkable intolerance towards mismatched sequences. Furthermore, it exhibits dual activation modes—responding to both DNA and RNA targets. The culmination of our research efforts has led to the development of the Hyper-Active-Verification Establishment (HAVE, 惠父). This innovation enables swift and precise *human papillomavirus* (HPV) diagnosis in clinical samples, providing a robust molecular diagnostic tool based on the Type I-A CRISPR-Cas3 system. Our findings contribute to understanding type I-A CRISPR-Cas3 system regulation and facilitate the creation of advanced diagnostic solutions with broad clinical applicability.

CRISPR-Cas immunity uses RNA-guided nucleases to target and degrade foreign nucleic acids in bacteria, archaea, and phage[1–6]. CRISPR-Cas systems are divided into Class 1 and Class 2 families depending on the copy number of comprised Cas protein[3,7,8]. Class 1 CRISPR–Cas system has multiple Cas proteins that form a crRNA-binding complex and function together to bind and cut the target[9–13]. Class 2 CRISPR-Cas system features a single, multi-domain crRNA-binding protein that is responsible for both recognition and cleavage of the target[3,11,14,15]. Due to the smaller size, Class 2 family Cas proteins have been widely deployed for revolutionary gene-editing tools and nucleic acid detection[16–26]. Although the application of the Class1 family may be limited by the factors such as multiple number of components, large size, and more complicated system, it also obtained extensive achievements in the field of gene editing and advancing the development of molecular diagnostic tools[6,16,23,27–34]

Type I CRISPR-Cas systems have garnered considerable attention due to their unique attributes and their potential applications in nucleic acid detection[35]. This system consists of six Cas proteins, comprising Cascade (Cas5, Cas6, Cas7, Cas8, Cas11), and the Cas3 nuclease (HD and HEL complex)[36]. Within the Type I-A system, Cas3 plays a pivotal role by forming the integral Cascade-Cas3 effector complex during the interference stage. Notably, when Cas3 operates as a standalone entity, it exhibits exceptionally robust single-stranded DNA (ssDNA) cleavage activity. However, upon binding to the Cascade complex, its conformation shifts to an inactive state. It's only when the Cascade-Cas3 complex encounters and opens the R-loop formation that Cas3's nuclease activity is reignited[37]. This resurgence leads to collateral, non-specific ssDNA cleavage, rendering it an attractive candidate for nucleic acid detection. Nonetheless, the high nuclease activity displayed by free Type I-A Cas3 presents a challenge in minimizing background noise and reducing false-positive outcomes. To tackle these challenges, our study unveils a novel Type I-A Cas3 variant that remains inactive until precisely triggered by the Cascade complex and the presence of target DNA. This Type I-A CRISPR-Cas3 system not only showcases an expanded protospacer adjacent motif (PAM) recognition capacity but also displays remarkable intolerance for mismatched sequences. Furthermore, it features dual activation modes, responsive to both DNA and RNA targets. This breakthrough significantly reduces the likelihood of false-positive results, ultimately elevating the precision and dependability of nucleic acid detection methodologies.

Accurate and sensitive detection of *human papillomavirus* (HPV) infections is of utmost importance for effective disease management and

prevention of HPV-associated malignancies[38,39]. The development of robust nucleic acid detection tools has significantly advanced the field of diagnostics, with the revolutionary CRISPR-Cas systems offering promising avenues for enhanced HPV detection[23]. Armed with this new variant of Type I-A Cas3, we proceeded to repurpose it to detect HPV infections. Leveraging the improved specificity and reduced background activity of this Cas3 variant, we developed a highly sensitive, rapid, and accurate nucleic acid detection assay specifically tailored for HPV. This innovative approach harbors tremendous potential to transform the landscape of current HPV diagnostics, presenting a swift, cost-effective, and dependable testing method crucial for effective patient management and HPV-associated disease prevention. It sheds light on the successful repurposing of this new Type I CRISPR-Cas3 for robust pathogen diagnosis.

## Results

### Type I-A Cas3 from *Thermococcus siculi* retains an inactive state when it stays alone

In our prior study, the type I-A CRISPR-Cas3 system from *Pyrococcus furiosus (Pfu)* displayed potent collateral ssDNA nuclease activity when Cascade-Cas3 encountered target DNA, forming a full R-loop[37]. This made it an attractive candidate for nucleic acid detection. However, when Cas3 operated in isolation, it exhibited high ssDNA nuclease (ssDNase) activity, posing a challenge for diagnostic applications due to potential background noise and false-positive results. To address this challenge, we conducted a homolog search for the *Pfu* Cas3 HD component in the JGI databank[40,41], identifying 88 conserved Cas3 HD homologs. Phylogenetic analysis revealed

their shared evolutionary origin (Fig. 1a). Upon annotating the gene cluster, we focused on two highly conserved homologs, *Pyrococcus Kukulkanii NCB100 (Pku)* and *Thermococcus siculi RG-20 (Tsi)*. Both strains exhibited a Type I-A CRISPR-Cas system adjacent to a Type I-B system, sharing the same CRISPR array and Cas1/Cas2 integrase system (Fig. 1b–d; Supplementary Fig. 1). Additionally, numerous accessory proteins were associated with these two CRISPR-Cas systems, offering insights into their defense coupling and a potential new accessory immunity pathway against viral invasion (Fig. 1b–d). Despite high sequence identity between *Pyrococcus Kukulkanii, Pyrococcus furiosus,* and *Thermococcus siculi* in protein alignment (Fig. 1e; Supplementary Fig. 1a–c), analysis of growth conditions revealed a distinct optimal temperature for *Thermococcus siculi* at 83 °C, compared to *Pyrococcus Kukulkanii*'s 105 °C and *Pyrococcus furiosus*' 100 °C (Fig. 1f). This observation raises the intriguing possibility that the Type I-A Cas3 from *Thermococcus siculi* may possess a distinctive ssDNase activity. All purified *Pfu*, *Pku*, and *Tsi* Cas3 complexes, encompassing both HEL and HD components and exhibiting a high degree of purity (Supplementary Fig. 1d). Subsequent ssDNA nuclease activity assays have revealed a distinct observation: solely *Tsi*Cas3 exhibits a complete lack of activity when challenged with the ssDNA substrate (Fig. 1g). Despite challenging the *Tsi* Cas3 nuclease to extremely high temperatures, it exhibited only minimal nuclease activity at 85 °C, a stark contrast to the activity observed in Pfu Cas3 (Supplementary Fig. 1e). These findings highlight the *Tsi* Type I-A system as a remarkably resilient candidate, distinguished by its negligible nuclease background. The potential mechanism will be further examined in subsequent discussions. In general, our findings pave the way for the

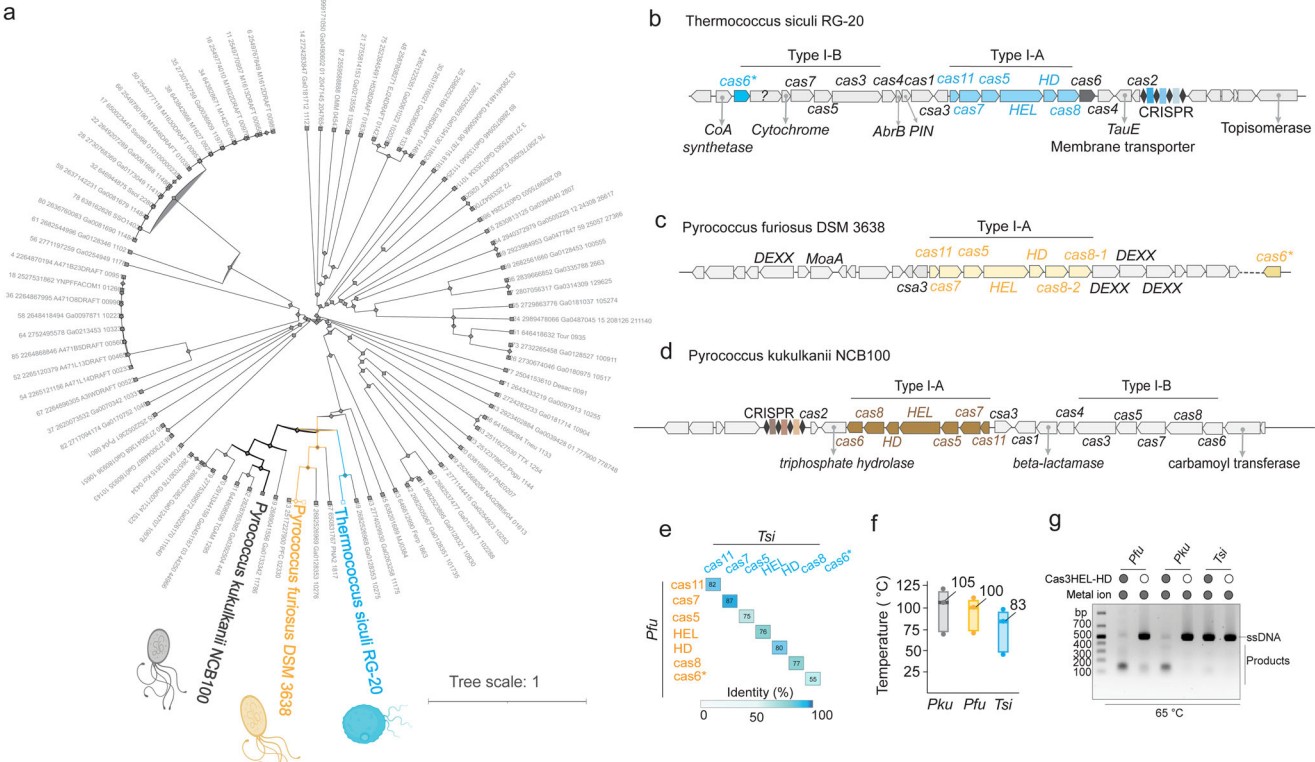

**Fig. 1 | Identifying a unique type I-A Cas3 variant devoid of background ssDNA nuclease activity in *Thermococcus siculi RG-20*. a** Phylogenetic tree depicting the relationships among Type I-A CRISPR-Cas3 systems, based on Cas3HD homologs. **b** Gene cluster representation of the Type I-A CRISPR-Cas3 system and its neighboring genes in *Thermococcus siculi RG-20*. Type I-A CRISPR-Cas3 system was highlighted in cyan color. **c** Gene cluster illustration of the Type I-A CRISPR-Cas3 system and its neighboring genes in *Pyrococcus furiosus DSM 3638*. Type I-A CRISPR-Cas3 system was highlighted in yellow color. **d** Gene cluster representation of the Type I-A CRISPR-Cas3 system and its neighboring genes in *Pyrococcus*

*Kukulkanii NCB100 (Pku)*, with the Type I-A CRISPR-Cas3 system highlighted in brown. **e** Chart Showcasing the percentage identity of Type I-A CRISPR-Cas3 components between *Pfu* and *Tsi*. It is important to note that the *Tsi*Cas6* included in the alignment originates from the neighboring Type I-B system, and the *Pfu*Cas6* is not part of the Type I-A gene cluster. **f** Survival conditions of *Pku, Pfu*, and *Tsi*, with the bar indicating the growth temperature range and the value indicating the optimal growth temperature. **g** Results of the single-stranded DNA (ssDNA) nuclease assay for Cas3 from *Pku, Pfu*, and *Tsi*.

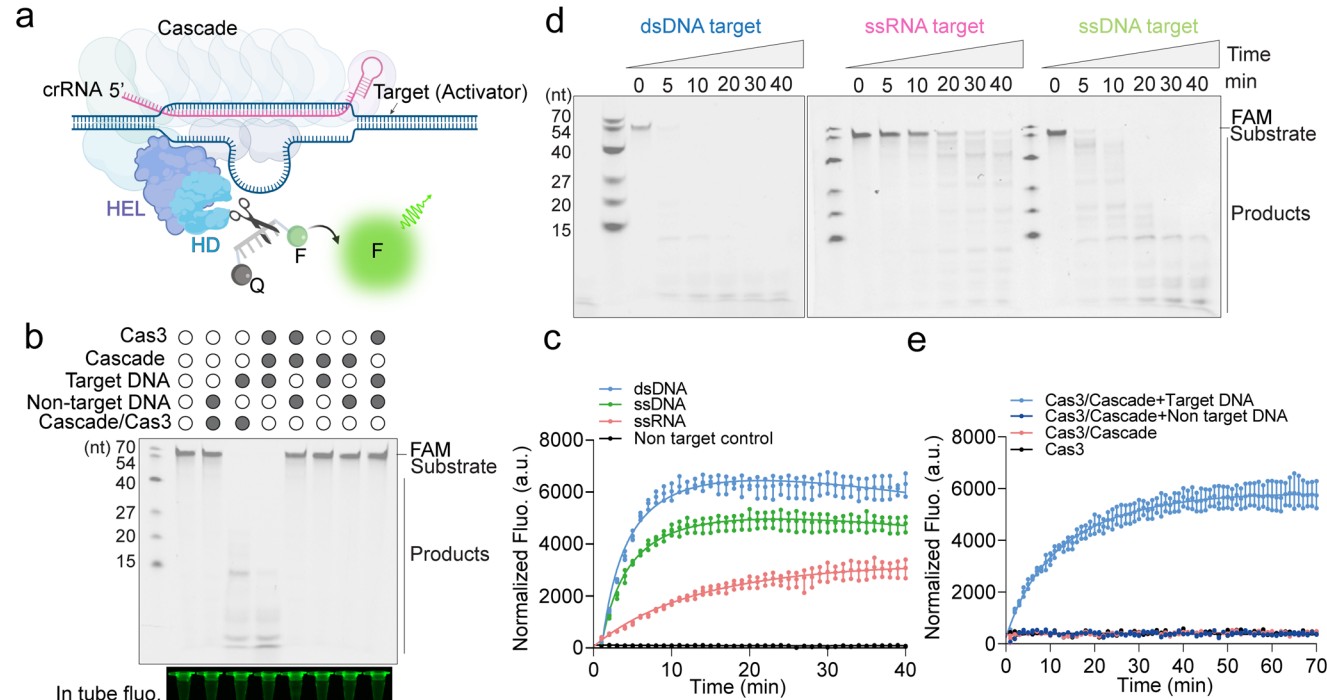

**Fig. 2 | Robust collateral ssDNA nuclease activity of the *Tsi* type I-A CRISPR-Cas3 complex triggered by both DNA- and RNA-target activators. a** Schematic representation of the *Tsi*Cascade-Cas3-mediated collateral ssDNA trans-cleavage activity. **b** Top: Denaturing-PAGE showcasing the trans-cleavage of *Tsi*Cascade-Cas3 on the FAM labeled ssDNA reporter, with both target and non-target DNA. Bottom: Illustration of component combinations that yield minimal background noise and a strong positive signal within the test tubes. The green arrow indicates the position of the reaction and fluorescence signal. **c** Real-time fluorescence monitoring assay for the detection of collateral trans-cleavage activity by *Tsi*Cascade-Cas3 nuclease on the F-Q single-stranded DNA reporter using target and non-target DNA activators. **d** Trans-cleavage assays of *Tsi*Cascade-Cas3 on FAM labeled ssDNA reporter, employing double-stranded DNA (dsDNA)-, single-stranded DNA (ssDNA)-, and single-stranded RNA (ssRNA)-target activators. Denaturing-PAGE reveals the time-dependent and robust ssDNA trans-cleavage activity of *Tsi*Cascade-Cas3 targeting dsDNA, ssRNA, and ssDNA activators. **e** Real-time fluorescence monitoring assay for the detection of collateral trans-cleavage of *Tsi*Cascade-Cas3 nuclease on the F-Q ssDNA reporter by targeting 20 nM dsDNA-, ssRNA-, ssDNA-, and non-target activators.

development of a solely Cas3 variant without ssDNase activity, addressing a critical limitation in diagnostic applications.

**_Tsi_ Type I-A CRISPR-Cas3 complex exhibits robust collateral ssDNA nuclease activity by both DNA- and RNA-target activators**

The subsequent challenge involves the acquisition of a *Tsi* type I-A Cascade complex comprising Cas5, Cas6, Cas7, Cas8, and Cas11 for conducting comprehensive biochemical investigations. Notably, Cas6a recognition and cleavage of the CRISPR repeat hinge on a non-stem loop structure inherent to the Type I-A system[42,43] (Fig. 2a, b). Considering that type I-A systems originate from hyperthermophiles, the straightforward transfer of the native gene cluster, including Cas6a which functions optimally at 70 °C[44,45], into *Escherichia coli (E. coli)* for expression and purification faces challenges due to the disparity with *E. coli*'s typical culture temperature of 37 °C. To surmount this challenge, we employed an approach by incorporating an *Eco-Tsi* chimeric *Ecoli*'s I-E system repeat along with *Tsi* I-A system 5′ tag (Supplementary Fig. S2c). This enables us to purify the complex by over-expressing the *Tsi*Cascade cluster and *Eco*Cas6e, which cleaves the chimeric repeat-5′ tag region. Subsequently, we achieved the *Tsi*Cascade-Cas3 complex by introducing Cas3 prior to gel-filtration, as outlined in Supplementary Fig. S2d–f. In pursuit of an optimal condition for collateral trans-cleavage activity on the FAM Fluorophore-Quencher (F-Q) labeled ssDNA reporter (Fig. 2a), we conducted a comprehensive analytical performance optimization by introducing target DNA with a 5′-CCC PAM motif, drawing inspiration from the homologous *Pfu* Type I-A system[46]. This effort yielded an ideal set of conditions, comprising an operating temperature of 85 °C (Supplementary Fig. S3a, b), 5 mM MgCl₂ (Supplementary Fig. S3c), omission of ATP (Supplementary Fig. S3d, e), and the utilization of a poly(d)

T F-Q ssDNA reporter (Supplementary Fig. S3f, g). In our denaturing-polyacrylamide gel electrophoresis (PAGE) gel analysis, we observed robust collateral trans-cleavage activity when Cascade-Cas3 encountered target DNA, whether through the co-mixing of Cascade and Cas3 components or by introducing the intact Cascade-Cas3 complex obtained through gel-filtration (Fig. 2b). It's noteworthy that Cas3 alone did not exhibit any cleavage activity on the FAM labeled ssDNA reporter, indicating the absence of background noise originating from this standalone nuclease component (Fig. 2b). Additionally, real-time fluorescence monitoring assays demonstrated evident collateral trans-cleavage activity by the *Tsi*-Cascade-Cas3 nuclease on the F-Q ssDNA reporter and this collateral activity was exclusively triggered by target activators only (Fig. 2c). To explore the potential involvement of different target molecules in activating *Tsi*Cascade-Cas3 collateral activity, we conducted collateral activity assays employing dsDNA-, ssRNA-, and ssDNA-target activators. The findings indicated that both dsDNA and ssDNA were capable of triggering *Tsi*-Cascade-Cas3's collateral trans-cleavage activity with a comparable level of effectiveness (Fig. 2d, e). Notably, the ssRNA-target activator also induced the trans-cleavage activity of *Tsi*Cascade-Cas3 (Fig. 2d, e). Although the ssRNA activator yielded a reduced effect compared to ssDNA and dsDNA (Fig. 2e, Supplementary Fig. S4a, b), these findings underscore the potential of RNA detection based on Type I-A systems, offering promise for applications such as SARS-COV19 or other RNA transcripts detection. Furthermore, when we annealed the ssRNA into dsRNA using a complementary RNA strand, Cascade-Cas3 did not display collateral activity with dsRNA, suggesting an inability to open dsRNA (Supplementary Fig. S4c, d). Accordingly, we conducted trans-cleavage assays employing dsDNA, ssRNA, and ssDNA target activators across a range of

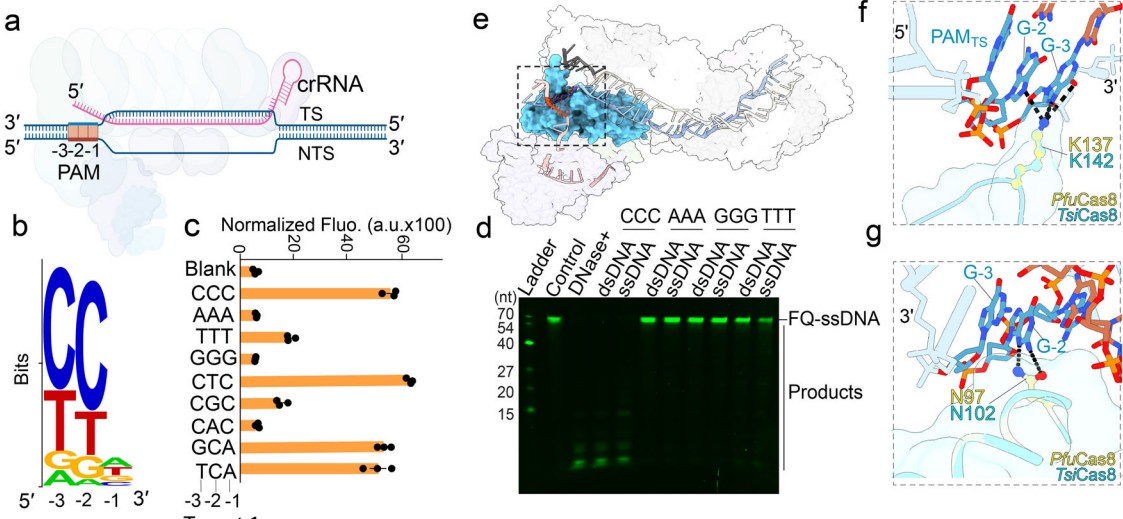

**Fig. 3 | Expanding PAM recognition in the type I-A CRISPR-Cas3 system.**
**a** Schematic representation of PAM recognition and R-loop formation within the
Type I-A Cascade-Cas3 system, with the PAM motif from the non-target strand
(NTS) highlighted in orange. **b** Bioinformatic prediction defines a 5′ - $Y_{-3}Y_{-2}N_{-1}$
PAM motif (where Y represents Cytosine or Thymine). Natural target identification
for native spacers in the *Tsi* Type I-A system was performed using CRISPRTarget.
**c** Fluorescence monitoring assay illustrating the collateral activity of *Tsi*Cascade-
Cas3 on the F-Q ssDNA reporter, with dsDNA target 1 featuring diverse PAM
sequences. **d** Denaturing-PAGE gel presenting the trans-cleavage activity of

*Tsi*Cascade-Cas3 on the FAM labeled ssDNA reporter, employing dsDNA targets
with varying poly-deoxyribonucleotide PAM sequences. **e** 3D structural model of
the *Tsi* Type I-A CRISPR-Cas3 complex bound to the dsDNA target. The PAM
recognition component, Cas8, is highlighted in cyan, and the PAM region is marked
in red. **f, g** Comparative structural models illustrating the structural basis of PAM
recognition by *pfu* and *Tsi* Cas8 components. *Tsi*Cas8 utilized identical residues to
interact with the PAM motif, exemplified by *Tsi*Cas8's use of K142 to contact G-3
and G-2, and N102 to engage G-2 in the PAM-complementary target-strand, mir-
roring the corresponding *Pfu*Cas8 residues, K137 and N97.

F-Q ssDNA reporter concentrations to analyze the dynamics of the cleavage
reaction (Supplementary Fig. S5a). Michaelis-Menten analyses unveiled
that the catalytic efficiency of the ssDNA activator (kcat/km) stood at
$6.50 \times 10^7 \, S^{-1} M^{-1}$, comparable to the dsDNA activator ($7.46 \times 10^7 \, S^{-1} M^{-1}$)
and was four-fold higher than that of the RNA activator ($1.60 \times 10^7 \, S^{-1} M^{-1}$)
(Supplementary Fig. S5b). Moreover, we conducted a sensitivity and spe-
cificity analysis for the Tsi Type I-A CRISPR-Cas3 system. Supplementary
Fig. S5c presents calibration plots for detecting three types of targets using
this system. It establishes the limit of detection (LOD) at 0.35 pM for
dsDNA (within a linear range of 1–100 pM and an $R^2$ of 0.9798), 0.95 pM for
ssDNA targets (across a 5–100 pM range with an $R^2$ of 0.9811), and 1.1 pM
for RNA targets (also within a 5–100 pM range and an $R^2$ of 0.9785).

### *Tsi* Type I-A CRISPR-Cas3 system exhibits expanded PAM recognition

We embarked on a quest to unveil the PAM signature within the *Tsi* Type
I-A CRISPR-Cas3 system. Our initial step involved identifying native
spacers within the *Tsi* Type I-A system using the CRISPRTarget tool[47].
The results yielded a distinctive 5′- $Y_{-3}Y_{-2}N_{-1}$ PAM motif, with "Y"
representing either Cytosine or Thymine (Fig. 3a, b). This finding implies
that *Tsi* Type I-A CRISPR-Cas3 can recognize remarkable 16 PAM
motifs, indicating an exceptionally broad PAM recognition spectrum,
akin to observations in the *Pfu* Type I-A system[48]. Next, we conducted
collateral activity assays to substantiate the PAM motifs' functionality.
To eliminate spacer DNA sequence bias, we generated two distinct tar-
gets, target 1 and target 2, each featuring varying DNA sequences within
the spacer region (Supplementary Fig. S6a). Subsequently, we performed
fluorescence monitoring assays to detect collateral activity on the F-Q
ssDNA reporter, employing dsDNA targets with diverse PAM motifs
(Supplementary Fig. S6a). The results illustrated a preference within the
*Tsi* Type I-A system for Cytosine and Thymine richness in the -3 and -2
positions of the PAM motif, independent of spacer DNA sequences
(Fig. 3c and Supplementary Fig. S6b). This preference was further cor-
roborated through denaturing PAGE gel assays, which indicated

*Tsi*Cascade-Cas3's inclination toward poly dC and poly dT (Fig. 3d).
Utilizing AlphaFold2, we predicted the *Tsi*Cas8 structure, revealing
striking structural similarity to *Pfu*Cas8 with a Cα-Root-Mean-Square
Deviation value of merely 0.1 Å (Supplementary Fig. S6c). Employing
this insight, we successfully docked *Tsi*Cas8 into the *Pfu*Cascade-Cas3/
R-loop formation structure with 5′- $C_{-3}C_{-2}C_{-1}$ PAM motif, thereby
obtaining a reasonable 3D model for in-depth analysis of the structural
basis of PAM recognition (Fig. 3e). Notably, *Tsi*Cas8 utilized identical
residues to interact with the PAM motif, exemplified by *Tsi*

Cas8's use of K142 to contact $G_{-3}$ and $G_{-2}$, and N102 to engage
$G_{-2}$ in the PAM-complementary target-strand, mirroring the corre-
sponding *Pfu*Cas8 residues, K137 and N97 (Fig. 3f, g, Supplementary
Fig. S6d). To further validate the PAM motif's versatility, we crafted a
comprehensive PAM library of plasmids encompassing 64 PAM
motifs. Subsequently, we conducted an Electrophoretic Mobility Shift
Assay (EMSA) and followed it with DNA sequencing to meticulously
map the PAM preference within the target DNA containing the PAM
library (Supplementary Fig. S7a–c). Our sequencing results unveiled
that *Tsi*Cascade exhibits a pronounced preference for Cytosine at the
-3 and -2 positions, particularly at the -2 site. Meanwhile, a slight
preference for Thymine was observed at the same positions. Intri-
guingly, at the -1 site, no discernible nucleotide preference was
defined from the sequencing data (Supplementary Fig. S7d, e). Fur-
thermore, we aimed to ascertain the similarity in PAM motif
requirements for ssRNA targeting compared to DNA. To this end, we
performed a biochemical assay to evaluate four polynucleotide PAM
motifs within ssRNA targets. The results consistently indicated that
only the CCC PAM motif is effective for RNA targeting, suggesting a
similar PAM recognition pattern between DNA and RNA targets
(Supplementary Fig. S6e). Collectively, our bioinformatics predic-
tions, collateral activity assays, and EMSA-sequencing assays provide
robust validation of the expanded PAM recognition activity inherent
to the *Tsi* Type I-A CRISPR-Cas3 system, thus widening the possi-
bility of target DNA selection in the gene-engineering.

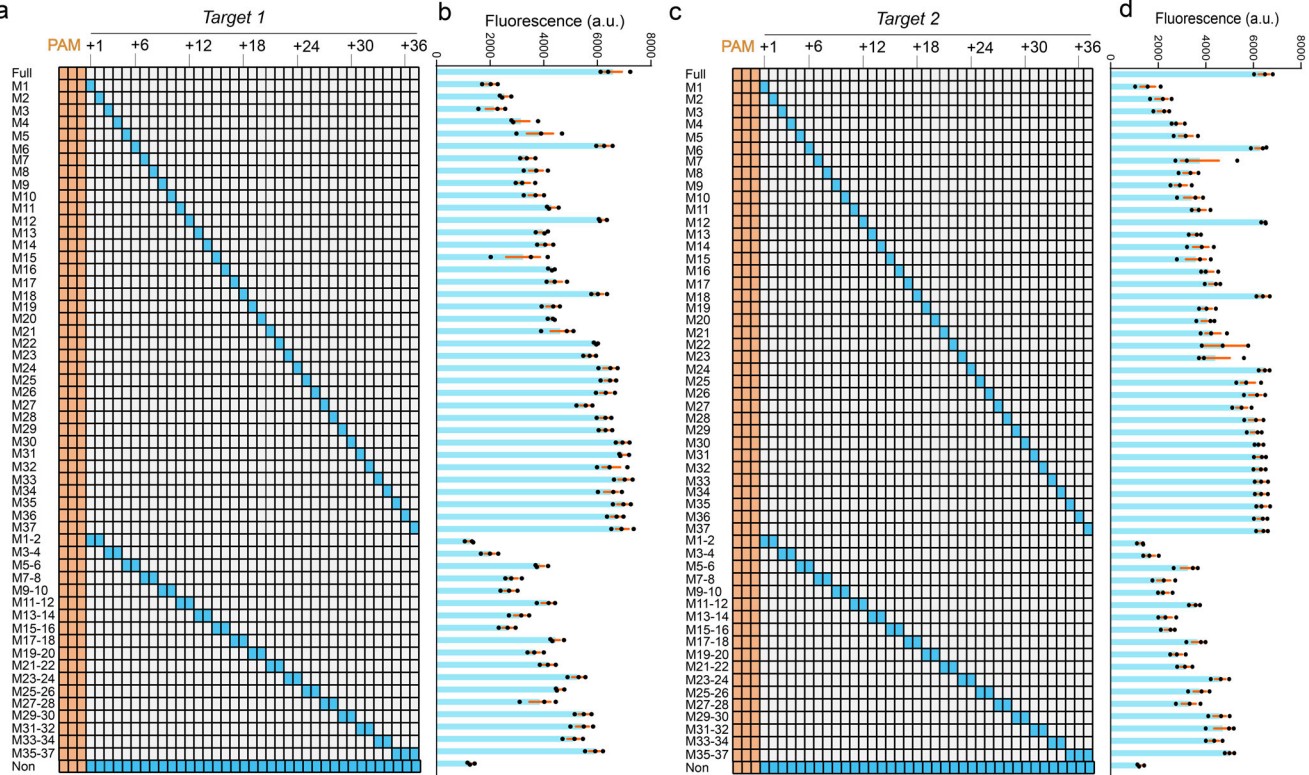

**Fig. 4 | Stringent mismatch-intolerance demonstrated by the *Tsi* type I-A CRISPR-Cas3 system. a** Paired Target 1 was strategically designed to contain mismatches at one (M1 to M37) or multiple (M1 ~ 2 to M35 ~ 37) sites within the guide RNA 1. The mismatched nucleotide was denoted by cyan boxes. **b** Evaluation of the relative trans-cleavage efficiency of Target 1 on the F-Q ssDNA reporter, measured through fluorescence monitoring assays. The columns represent the average values derived from three independent quantitative datasets, detailed in the corresponding column. **c** Paired Target 2 was strategically designed to contain mismatches at one (M1 to M37) or multiple (M1 ~ 2 to M35 ~ 37) sites within the guide RNA 1. The mismatched nucleotide was denoted by cyan boxes. **d** Evaluation of the relative trans-cleavage efficiency of Target 2 on the F-Q ssDNA reporter, measured through fluorescence monitoring assays. The columns represent the average values derived from three independent quantitative datasets, detailed in the corresponding column.

## Stringent mismatch-intolerance in the *Tsi* type I-A CRISPR-Cas3 System

Following our exploration of PAM recognition activity, we sought to elucidate the mismatch tolerance exhibited by the *Tsi* Type I-A CRISPR-Cas3 system. To eliminate potential bias arising from spacer DNA sequences, we conducted comprehensive evaluation assays using two distinct target DNA sequences, each sharing the same 5′- $C_{-3}C_{-2}C_{-1}$ PAM motif. In a meticulous evaluation of mismatch tolerance, for each target DNA, we generated a total of 55 target DNAs, comprising 37 featuring a single mismatch and 18 with multiple mismatches (Fig. 4a, c). Subsequently, we monitored the fluorescence signals to assess the collateral trans-cleavage activity on the F-Q ssDNA reporter, conducting assays with *Tsi*Cascade-Cas3 and a total of 110 mismatched target DNAs (Fig. 4a, c). Our findings reveal that mismatches within the PAM-proximal seed region significantly impair trans-cleavage activity, except mismatches at every 6th position showed no impact, whereas mismatches positioned distally from the PAM are notably well tolerated, particularly beyond 24 bases into the paired target DNAs, leading to nearly full-matched collateral efficiencies (Fig. 4b, d). However, when clusters of mismatches were distributed across double and triple sites, collateral activity experienced a dramatic reduction across the initial 16 bases of paired target DNAs tested (Fig. 4b, d). Moreover, The 6th base of each crRNA segment, being sequestered within Cas7 and unable to base pair with the target, means that mutations at these positions (e.g., the 6th, 12th, 18th, 24th, etc.) do not affect the system's collateral activity. This observation aligns with previous findings related to the Type I-E system[49]. This comprehensive assessment of mismatched target DNAs suggests that effective targetable sites are expected to occur approximately every ~24 base

pairs within target DNA selection, with potential off-target sites featuring optimal interspacer distances remaining a rarity. In sum, this thorough characterization of mismatch tolerance within the *Tsi* Type I-A CRISPR-Cas3 system not only provides valuable guidance for target DNA selection in in vivo gene-editing applications but also establishes a designable standard for in vitro nucleic acid detection.

## Development of a portable detector for rapid nucleic acid detection using the *Tsi* type I-A CRISPR-Cas3 system

Our next objective was to transform our Type I-A CRISPR-Cas3 system into a Point-of-Care (POC) detection tool, addressing the pressing need for a portable device. Our aim was to create a user-friendly, cost-effective, and portable solution that would eliminate the requirement for specialized laboratory equipment, while efficiently interpreting the fluorescent signals generated by our Type I-A CRISPR-Cas3 system. To achieve this, we meticulously designed and developed a compact apparatus with a minimal size and weight (Fig. 5a). A comprehensive depiction of its core components, including both three-dimensional and real-device images, is presented in engineering drawings and a sectional view (Fig. 5b, c). This device primarily encompasses an optical module, a temperature control module, and a user-friendly touchscreen interface. Within the optical module featuring eight sample channels, a 470-nm light-emitting diode serves as the fluorescence excitation module, directing light toward the tube bottom. Additionally, an emission filter (525 nm) and a data processing system are situated at the lateral position for precise analysis of fluorescence intensity (Fig. 5d). Despite its smaller size compared to expensive and bulky quantitative real-time PCR devices, our device maintains a temperature control range from

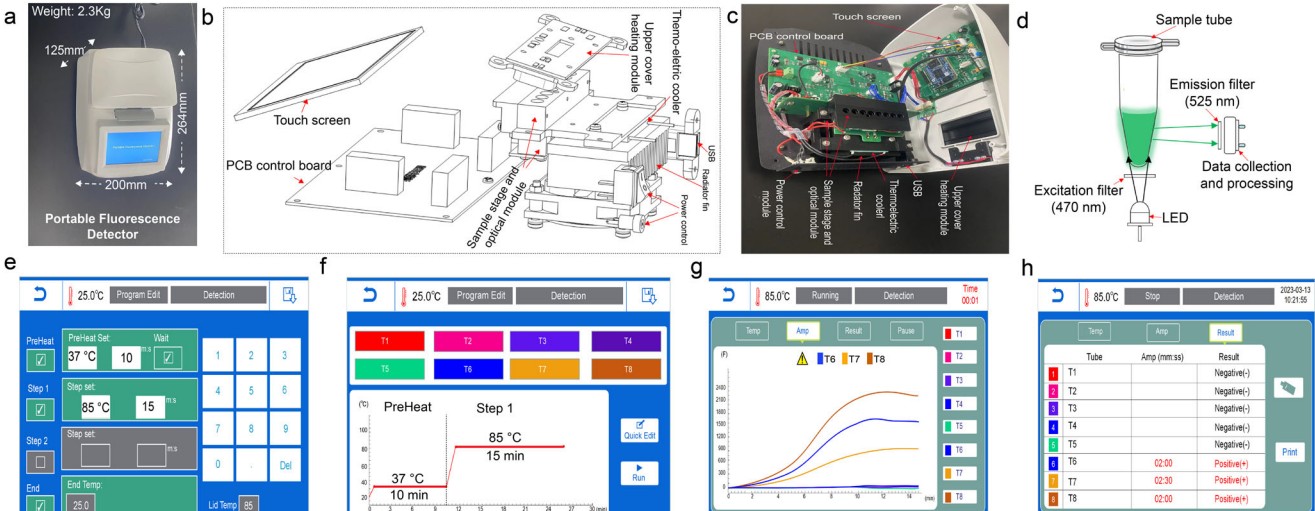

**Fig. 5 | Engineering a portable detector enables the rapid establishment of nucleic acid detection based on *Tsi* type I-A CRISPR-Cas3 system. a** An illustration portraying the portable fluorescence detector, emphasizing its compact dimensions measuring 200 × 264 × 125 mm, and its capability to simultaneously analyze up to 8 samples. Technical diagrams illustrating the integral components of the portable detector, are presented in both engineering drawing (**b**) and a sectional view (**c**). **d** Schematic depiction elucidating the mechanism of fluorescence data acquisition employed in the device. **e** Diagram outlining the control panel interface for setting up the nucleic acid detection reaction. **f** Overview of the nucleic acid detection program leveraging the *Tsi* Type I-A CRISPR-Cas system. **g** Real-time fluorescence signal monitoring for the samples undergoing detection. The real-time curve is displayed on the control panel, with positive results indicated at the top. **h** A summarized report detailing the detection results, with positive samples prominently marked in red and accompanied by the associated cut-off time of the positive signal, enabling assessment of the target molecule's concentration.

25 °C to 100 °C, perfectly suited for the requirements of our platform at 85 °C. We conducted evaluations of the device's practical detection capabilities using CRISPR-Cas3 for simultaneous detection of eight samples, with a detailed demonstration provided in Supplementary Movie 1. The detection program consists of two key phases: an initial preheat process facilitating RPA amplification (10 min) at 37 °C, followed by the execution of the CRISPR-Cas3-mediated collateral reaction procedure (15 min) at 85 °C, during which real-time fluorescence signals are continuously monitored (Fig. 5e, f). Notably, we incorporated an attention symbol that automatically appears upon the detection of positive results (Fig. 5g). In this demonstration, tubes 6, 7, and 8 were individually injected with 50, 20, and 100 pM dsDNA activators, effectively showcasing the instrument's competence in the detection (Fig. 5g). Following the entire detection process, we effortlessly obtained a summarized report detailing the detection results. Positive samples were conspicuously highlighted in red, along with the corresponding cut-off time of the positive signal, facilitating accurate assessment of the target molecule's concentration (Fig. 5h). This engineering feat exemplifies the potential application of our Type I-A CRISPR-Cas3 system for Point of Care (POC) applications.

### Repurpose *Tsi* type I-A CRISPR-Cas3 for a robust diagnosis of HPV clinical samples

Our journey towards harnessing the potential of the *Tsi* Type I-A CRISPR-Cas3 system led us to develop a diagnostic platform, capitalizing on the capabilities of a portable detector. Our initial target was the HPV, a leading cause of cervical cancer, primarily driven by high-risk HPV subtypes such as 16/18/45/52 (Supplementary Fig. S8a). Presently, the gold standard for precise identification of HPV genomic DNA, specifically within the L1 gene, is the PCR-Reverse Dot Blot (PCR-RDB) assay (Supplementary Fig. S8b, c). However, PCR-RDB necessitates specialized equipment, skilled operators, and substantial processing time (Supplementary Fig. S8d, e). This creates a compelling demand for a versatile, rapid, and portable biosensor, capable of addressing pressing healthcare and security needs. In response, we developed a highly active system and aptly named it "Hyper-Active-Verification Establishment" or HAVE, pronounced as /hei:v/. This name pays homage to Huifu (惠父), a courtesy name of an ancient Chinese forensic scientist Song Ci, widely revered as the father of world legal medicine. The HAVE assay workflow, depicted in Fig. 6a, achieves a sample-to-answer time of approximately 35 min. To discriminate between prevalent high-pathogenicity HPV types 16 and 18, we meticulously designed guide crRNAs, namely Guide 16 and Guide 18, based on L1 gene fragments (Fig. 6b, c). Leveraging distinct PAM motif sequences (Fig. 6b, c), these guides enable subtype discrimination. Our initial validation process entailed identifying HPV 16 and 18 DNA in 30 human anal swab samples using the established gold-standard PCR-RDB method, as detailed in Supplementary Figs. S8d–f. The results for subtypes 16 and 18 are presented in Figure 6d. When applying the HAVE platform to these 30 samples, we observed a 100% concordance with the PCR-RDB assay in detecting subtype 16. For subtype 18, our method achieved a 93.33% accuracy rate, which accounts for one false-positive case, as detailed in Fig. 6e. This groundbreaking achievement not only repurposes the *Tsi* Type I-A CRISPR-Cas3 system but also heralds a new era of robust and efficient HPV clinical sample diagnosis. With the development of this portable, rapid, and accurate diagnostic tool, we hope to significantly impact healthcare and global security by offering an accessible and reliable solution for HPV detection, ultimately contributing to the prevention and early treatment of HPV-associated diseases, particularly cervical cancer. Furthermore, the Type I-A CRISPR-Cas3 system is shown to effectively distinguish between HPV 16 and HPV 18, as opposed to three other interfering agents (Fig. 6f)

## Discussion

In this study, we explored the potential of Type I-A CRISPR-Cas3 for nucleic acid detection, focusing on its hyperactive ssDNA nuclease activity. Our findings not only shed light on the regulatory mechanisms of Type I-A CRISPR-Cas3 but also culminated in the development of the Hyper-Active-Verification Establishment (HAVE) for swift and precise HPV diagnosis in clinical samples.

One of the pivotal accomplishments of this study lies in the discovery of a novel Type I-A Cas3 variant derived from *Thermococcus siculi*, showcasing a distinct regulatory mechanism. In contrast to the hyperactive standalone Cas3 nucleases commonly observed in many Type I systems[35,37,50–52], this particular Cas3 variant remains in an auto-

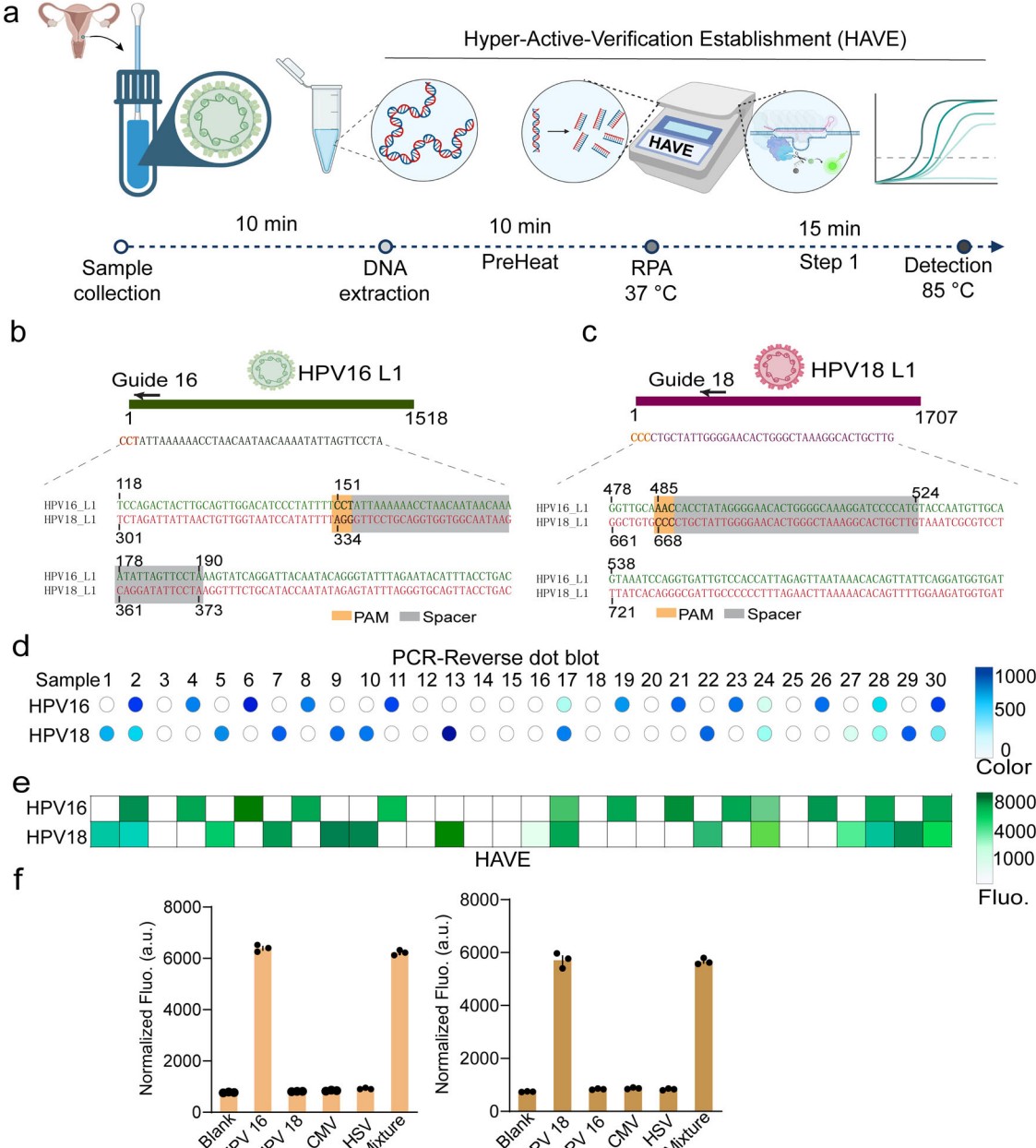

**Fig. 6 | Repurposing *Tsi* type I-A CRISPR-Cas3 for Hyper-Active-Verification Establishment (HAVE) to enable robust diagnosis of HPV clinical samples. a** A schematic diagram illustrating the principles and workflow of Hyper-Active-Verification Establishment (HAVE), encompassing DNA extraction (10 min), RPA amplification (10 min), and *Tsi*Cascade-Cas3 mediated trans-cleavage for the release of fluorescence signals (15 min). Here we used the BioRender to make this figure. **b**, **c** The scheme illustrates the design of guide crRNAs Guide 16 and Guide 18, which are constructed using L1 gene fragments. These guides utilize distinct PAM motif sequences to discriminate between HPV subtypes 16 (**b**) and 18 (**c**). **d** A summary of diagnostic outcomes for clinical samples obtained from 30 patients, including 22 patients with HPV16 and HPV18 virus infections and 8 uninfected patients, assessed using PCR-Reverse dot blot assay. **e** A summary of diagnostic outcomes for the same clinical samples as in (**b**), utilizing fluorescence signal intensity analysis with the HAVE platform. Specificity Assessment of the Type I-A CRISPR-Cas3 System. **f** Demonstrates the system's specificity for HPV 16 (left) and HPV 18 (right), distinguishing it from cytomegalovirus (CMV), herpes simplex virus (HSV), and HPV variants. Error bars represent the standard deviation (SD) for $n = 3$ measurements.

inhibited state until it is activated by the presence of the Cascade complex and the formation of R-loops. This auto-inhibition mechanism constitutes a remarkable advancement within the realm of CRISPR-Cas systems, addressing a significant challenge associated with the hyper-active ssDNA nuclease activity of Cas3. This hyperactivity often introduces undesirable noise and false-positive signals in nucleic acid detection assays. Our in-depth characterization and harnessing of this auto-inhibition mechanism lay the cornerstone for the development of more precise and reliable CRISPR-Cas3-based diagnostic tools. A

compelling question arises regarding why TsiCas3 remains auto-inhibited when it operates independently, given the high degree of amino acid similarity between *Pfu*Cas3 and *Tsi*Cas3. To address this query, we turned to structural predictions using AlphaFold2. While we had previously determined the structures of *Pfu*Cas3 and Cascade, we sought to minimize potential biases arising from different structural resources. Therefore, we employed AlphaFold2 to generate a structure model for the *Pfu*Cas3 complex. This approach allowed us to obtain structural information from a uniform source. Intriguingly, our

structural analysis uncovered a notable difference: residue K218 in *Pfu*Cas3 was correspondingly replaced by E217 in *Tsi*Cas3. K218 possesses a higher affinity for contacting the DNA backbone besides the HD nuclease pocket, whereas E217 seems to repel the DNA backbone, potentially due to electrostatic repulsion (Supplementary Figs. S1e, S9a). Either glutamic acid or arginine here may serve as a "gate residue" to modulate the nuclease activity when Cas3 is in an isolated state. Furthermore, we observed that the C-terminal tail of *Tsi*Cas3HD is positioned closer to the active site compared to its counterpart in *Pfu*Cas3 (Supplementary Fig. S9a). This proximity restricts the access of *Tsi*Cas3HD to the DNA substrate, rendering it more stringent in interacting with DNA. Moreover, the inclusion of Cas3HEL in our analysis revealed additional insights. In *Pfu*Cas3HEL, we observed the clustering of positive residues, whereas *Tsi*Cas3HEL displayed a clustering of negatively charged residues (Supplementary Fig. S9b–d). Although the corresponding site in Pku Cas3HD is also E217, PkuCas3 exhibits strong ssDNase activity, suggesting that auto-inhibition involves a complex regulatory mechanism and cannot be solely determined by this site. Interestingly, when we introduced an E217K mutation in TsiCas3HD, this mutant displayed moderate ssDNase activity (Supplementary Fig. S9e), highlighting the significance of this site for the HD domain's engagement with ssDNA substrates in the Tsi variant. This disparity further elucidates the stringent auto-inhibition observed in *Tsi*Cas3 when it functions independently. This understanding of the regulatory mechanisms governing *Tsi*Cas3 not only enriches our knowledge of Type I CRISPR-Cas3 systems but also underscores the significance of harnessing such regulatory mechanisms for the development of more precise and reliable nucleic acid detection tools.

Our work also revealed the expanded PAM recognition capabilities of the Type I-A CRISPR-Cas3 system. PAM sequences are critical for target recognition and binding by CRISPR-Cas systems. The ability of this Type I-A system to recognize and cleave DNA with a broader range of PAM motifs, including those rich in cytosine (C) and thymine (T), is a significant finding. This expanded PAM recognition activity provides researchers and diagnosticians with a wider range of choices when designing target DNA sequences. It enhances the versatility of the system for nucleic acid detection and gene editing applications.

Another notable feature of the Type I-A CRISPR-Cas3 system demonstrated in this study is its exceptional mismatch tolerance. We systematically evaluated the system's ability to tolerate single and multiple mismatches in target DNA sequences. Our results showed that mismatches within the PAM-proximal seed region were highly deleterious to transcleavage activity, whereas mismatches distal from the PAM were well tolerated. This insight into the system's mismatch tolerance provides guidance for designing highly specific detection assays and minimizing false-positive results.

HPV infections, particularly those involving high-risk types such as HPV16 and HPV18, pose a substantial public health challenge, necessitating precise detection methods for effective disease management. Presently employed diagnostic techniques, such as PCR-Reverse Dot Blot (RDB), while recognized for their accuracy, come with certain drawbacks, including the requirement for specialized equipment, skilled personnel, and protracted turnaround times. In response to these challenges, we have devised an exceptionally active diagnostic system, aptly christened the "Hyper-Active-Verification Establishment" or HAVE, pronounced as /heiːv/. The selection of this moniker is a homage to Huifu (惠父), originally known as Song Ci, an ancient Chinese forensic scientist revered worldwide as the progenitor of word legal medicine. Given that our HAVE platform is meticulously designed for the purpose of diagnosis and detection, we found it fitting to adopt the courtesy name of this venerable figure in the nomenclature of our system. Our results unequivocally demonstrate that the HAVE platform achieves a remarkable 100% concordance rate with PCR-RDB, affirming its accuracy and unwavering reliability in the realm of HPV diagnosis. This innovative approach not only streamlines and expedites HPV testing but also holds the promise of adaptability for the diagnosis of

other pathogens and nucleic acid targets. The HAVE platform, therefore, marks a significant stride toward democratizing molecular diagnostics, particularly in resource-constrained settings where expeditious and dependable diagnostic tools are of paramount importance.

What's more, we conducted a comprehensive evaluation of the transcleavage activity of *Tsi*Cas3/Cascade in comparison to previously established CRISPR/Cas-based diagnostic systems (Supplementary Fig. S10a). For this comparison, we selected a Cas12a homolog from *Lachnospiraceae bacterium* known for its robust ssDNA cleavage activity. Analysis of the fluorescence dynamics revealed that the RNA-guided double-stranded DNA (dsDNA) binding target activator used in our study reached a plateau more rapidly than the CRISPR/Cas12a system when subjected to the same concentration of dsDNA activator at their respective optimal activity temperatures (Supplementary Fig. S10c, d). This observation suggests a comparable or even a slight shorter timeframe for nucleic acid diagnosis using the collateral cleavage of ssDNA-FQ property, thus highlighting the potential for expedited diagnostic processes. Additionally, we conducted tolerance experiments involving varying SDS concentrations and pH ranges to compare the resilience of the *Tsi*Cas3/Cascade system with the CRISPR/Cas12 system. Significantly, we observed that the TsiCas3/Cascade system displayed remarkable resilience to elevated SDS concentrations when compared to the CRISPR/Cas12 system (Supplementary Fig. S10e, f). This heightened tolerance can be attributed to the origin of *Tsi*Cascade/Cas3 from hyperthermophiles known for their exceptional thermal stability. Regarding pH conditions, the *Tsi*Cas3/Cascade system displayed a preference for acidic environments, while the CRISPR/Cas12 system thrived in alkaline conditions. These distinctions may be attributed to the origins of the *Tsi*Cas3/Cascade system, which hails from hyperthermophilic bacteria known for their preference for extreme conditions.

The development of universal detection techniques for the amplification-free recognition of DNA and RNA targets is critically important. Nevertheless, the application of the Type I-A CRISPR-Cas3 system in our study has shown its capability for target detection only at sub-picomolar concentrations. This limitation restricts its utility in contexts where detecting RNA targets at attomolar levels is necessary. Consequently, there is a compelling need for additional investigation to enhance this technology. This may include modifications to the Cas3 protein to improve its collateral activity or the integration with a droplet microfluidics approach for digital quantification of single-molecule targets.

In conclusion, our study advances our understanding of Type I-A CRISPR-Cas3 systems, introduces a novel regulatory mechanism, and presents a transformative solution for HPV diagnosis through the HAVE platform (Supplementary Video 1). This work opens new possibilities for the development of molecular diagnostic tools with broad clinical applicability, potentially revolutionizing disease detection and management.

## Methods
### Oligonucleotides
All the sequences (HPLC purified) used in this study were shown in Supplementary Data 1 document and purchased from General Biol (Anhui) Co., Ltd (Chuzhou, China). All oligonucleotides were dissolved in TE buffer at a concentration of 50 μM and stored at −20 °C.

### Plasmids construction and purification of TsiCascade-Cas3 proteins complex
The expression and purification of Cascade and Cas3 proteins followed the protocol outlined in our prior work (Hu et al., Mol. Cell., 2022, 82, 2754), with some adjustments for optimization. In brief, co-transformation of pCDFDuet-Twin-Strep-Cas8a (StrR), pETDuet-Cas11-Cas7-Cas5 (AmpR), and pcrRNA-Cas6e-chimeric crRNA (KanR) was performed using *E. coli* BL21 (DE3) competent cells. The bacterial culture was cultivated in 6 liters of LB medium at 37 °C, induced during the mid-log phase (OD600 ~ 0.6) with 0.5 mM IPTG, and subsequently shifted to 20 °C for overnight protein expression. The harvested cells were resuspended in buffer A (comprising 50 mM HEPES pH

https://doi.org/10.1038/s42003-024-06537-3 **Article**

7.5, 300 mM NaCl, 10% glycerol, and 2 mM β-ME), followed by cell lysis through sonication and clarification by centrifugation at 12,000 rpm for 50 min at 4 °C. The resulting supernatant was loaded onto a pre-equilibrated 4 mL streptavidin column (Twin-strep purification). Following a wash with 50 mL of buffer A, protein elution was carried out using 20 mL of buffer B (consisting of 50 mM HEPES pH 7.5, 300 mM NaCl, 10% glycerol, and 2.5 mM desthiobiotin). Further purification was accomplished through Superdex 200 16/60 size-exclusion chromatography (GE Healthcare) in buffer C (containing 10 mM HEPES pH 7.5 and 300 mM NaCl). The purified samples were flash-frozen and stored at −80 °C for future use.

The plasmids, pRSFDuet-Cas3HD, and Cas3HEL (KanR), were separately introduced into E. coli BL21 (DE3) competent cells, and their expression followed the previously described procedure. Post-expression, cells were harvested, and their lysis was carried out via sonication in 80 mL of buffer A comprising 50 mM HEPES at pH 7.5, 500 mM NaCl, 20 mM imidazole, 10% glycerol, and 2 mM β-ME. Following centrifugation, the resultant supernatant was loaded onto a pre-equilibrated 4 mL Ni-NTA column. Subsequent to a 100 mL wash with buffer A, protein elution was performed with 20 mL of buffer B, consisting of 50 mM HEPES at pH 7.5, 500 mM NaCl, 10% glycerol, 300 mM imidazole, and 2 mM β-ME. Further purification was accomplished through Superdex 200 16/60 size-exclusion chromatography (GE Healthcare) in buffer C, containing 10 mM HEPES at pH 7.5 and 300 mM NaCl. The purified samples were flash-frozen and stored at −80 °C for future use.

The purification of LbaCas12a was accomplished through a meticulous process. The 6XHis-MBP-TEVhuLbCpf1 plasmid (Addgene: 90096) underwent transformation into Escherichia coli Rosetta2 (DE3) cells, followed by selection of colonies and cultivation in Terrific Broth at 37 °C until the OD600 reached 0.6. Induction of protein expression was achieved through the addition of IPTG at 20 °C for 18 h. The resulting cell pellet was resuspended in lysis buffer (500 mM NaCl, 50 mM Hepes pH 7.5, 10% glycerol) and subjected to sonication for cell lysis. Subsequent centrifugation at 12,000 rpm for 50 min at 4 °C facilitated the collection of the soluble fraction, which was subsequently purified using a nickel column. Eluted fractions were then treated with Tobacco etch viruses protease to remove the 6His-MBP label, with the digestion taking place at 4 °C overnight. Following this step, another purification round involving a heparin column was carried out. Finally, the protein underwent overnight dialysis and was stored at −80 °C for future use.

### Processing of clinical sample
In this study, human cervical cell specimens ($n = 30$) from the International Maternity & Child Health Hospital, School of Medicine, Shanghai Jiao Tong University, was collected after written informed consent with approval from the MCHH Committee on Human Research (ZH2018QNA37). All ethical regulations relevant to human research participants were followed. Before inclusion in our analysis, these specimens underwent initial screening for HPV infection in the hospital's clinical laboratory. This screening employed PCR-RDB Dot Blot assays provided by Yaneng BIOscience in Shenzhen, China. At the meantime, we proceeded with the application of the HAVE platform to these specimens. This platform, designed for rapid and accurate HPV diagnosis, allowed us to assess its performance on real clinical samples. This step aimed to validate the platform's effectiveness and reliability in a clinical context, particularly for the detection of HPV, a crucial aspect of women's health.

### Fluorescence monitoring assay
The fluorescence detection assays employing *Tsi*Cascade-Cas3 were conducted using single-stranded oligonucleotide fluorophore-quencher (F-Q) reporters. In a 10 μL reaction solution, we combined 2 μL of *Tsi*Cascade-Cas3 at a final concentration of 200 nM, 0.5 μL of F-Q labeled ssDNA reporter at a final concentration of 20 nM, 1 μL of the target activator at a vary concentration, and 6.5 μL of reaction buffer (comprising 50 mM Tris-HCl pH 8.0, 100 mM KCl, 5 mM MgCl₂, and 5% glycerol). This reaction

mixture was subjected to incubation at 85 °C for 15 min to facilitate the reaction. Subsequently, the resulting fluorescence signal was detected and recorded using a Synergy H1 instrument. This assay configuration allowed us to assess the collateral trans-cleavage activity of *Tsi*Cascade-Cas3 efficiently. The F-Q reporters, in conjunction with the specific reaction conditions, enabled us to monitor and quantify the fluorescence changes associated with the activation of *Tsi*Cascade-Cas3 by the target activator, providing valuable insights into the system's nucleic acid detection capabilities.

### Denaturing PAGE
Denaturing PAGE served as a pivotal method for the comprehensive characterization of the cleavage products derived from the FAM-labeled ssDNA substrate. To initiate the reaction, 2 μL of TsiCascade-Cas3 was added, reaching a final concentration of 200 nM, alongside 0.5 μL of the FAM-labeled ssDNA, which was adjusted to a final concentration of 20 nM. To activate the reaction, 1 μL of the target activator was introduced, followed by the addition of 6.5 μL of reaction buffer (comprising 50 mM Tris-HCl at pH 8.0, 100 mM KCl, 5 mM MgCl₂, and 5% glycerol) into the 10 μL total reaction volume. The reaction mixture was maintained at 85 °C for a duration of 15 min. Subsequently, to halt the reaction and ensure accurate analysis, 10 μL of 95% formamine was mixed with the reaction buffer and subjected to a temperature of 95 °C for 3 min. For the electrophoretic separation of the reaction products, a 20% denaturing PAGE gel containing 7 M urea was utilized. Electrophoresis was carried out at a voltage of 300 V for a period of 70 min using the Mini-PROTEAN Tetra Cell system (Bio-Rad) in 0.5 X TBE buffer. The resulting gel was visualized and documented using the Gel Doc XR system (Bio-Rad Company, USA).

### Structural model analysis
In this study, we conducted an extensive analysis of the structural models to understand the molecular architecture and interactions within our target systems. A key component of our analysis is presented in Fig. 3e, where the predicted model of *Tsi*Cas8 was constructed using the advanced predictive capabilities of AlphaFold2[53]. Following the generation of the *Tsi*Cas8 model, we integrated this structure with existing Protein Data Bank data (PDB ID: 7TRA) using molecular docking techniques. The docking process was followed by a comprehensive analysis using Chimera X software[54]. Additionally, other figures in the manuscript, specifically 3 g and 4 f, were also generated using Chimera X.

### Michaelis-Menton kinetic study
The Michaelis-Menten analysis was conducted following the methodology outlined in a previously published reference (Xing et al., Adv. Sci., 2020, 7, 1903661[55]), with some adaptations. Various concentrations of F-Q labeled ssDNA substrate (ranging with 0.001, 0.01, 0.1, 0.2, 0.5, 1, 2, 5, and 10 μM) were subjected to incubation with 100 nM *Tsi*Cascade-Cas3 complex and 10 nM activators (comprising dsDNA, ssDNA, and ssRNA). The reaction progress was monitored in real-time using fluorescence measurements. To determine the Michaelis constant (Km) of *Tsi*Cascade-Cas3, we plotted the initial velocity (V0) against substrate concentration ([S]). The Michaelis-Menten equation, $V0 = (Vmax[S])/(Km + [S])$, where Vmax represents the maximum reaction rate, was employed. The turnover number (kcat) of *Tsi*Cascade-Cas3 was calculated as kcat = Vmax/Et, with Et representing the concentration of TsiCascade-Cas3 (fixed at 100 nM). Finally, the catalytic efficiency was determined by evaluating kcat/Km. This analysis allowed us to gain insights into the enzyme kinetics and efficiency of TsiCascade-Cas3 in the context of various substrates and activators.

### PAM library generation
The creation of a diverse PAM library involved the synthesis of two complementary oligonucleotides, PF-Mix-PAM and PR-Mix-PAM, obtained from Sangon Biotech. These oligonucleotides were designed to carry a 3 nt PAM sequence (NNN) at the 5' end of the protospacer sequence. Additionally, BamHI and XhoI cut sites were thoughtfully included, flanking the

entire sequence. After ensuring the 5'-OH phosphorylation and the annealing of these oligonucleotides, they were seamlessly ligated into the pET-28a vector using T4 Fast ligase. Subsequently, the ligation products were introduced into DH5α for transformation. The resulting PAM library impressively encompassed a repertoire of 64 potential PAM sequences. The coverage and accuracy of these sequences were thoughtfully verified through Sanger sequencing, performed with meticulous care by Sangon Biotech.

## PAM determination

The elucidation of PAM preferences within the Type I-A system was an intricate process. To commence, 161 bp double-stranded DNAs were diligently isolated from the PAM library through a judiciously planned PCR reaction, employing PF-161 and PR-161 as primers. Another noteworthy PCR amplification ensued, this time featuring PF-6-FAM and PR-161 primers, thus engendering 5'-6-FAM labeled PCR products. The ensuing 161 bp 6-FAM-PAM library DNA molecules, valuing at 320 nM, were subjected to a rigorous incubation with varying concentrations of the *Tsi*-Cascade-Cas3 complex (ranging from 0 to 2000 nM). This meticulous procedure unfolded in a buffer comprising 20 mM HEPES (pH 7.5) and 100 mM NaCl, all maintained at a temperature of 37 °C for an hour. The samples that emerged from this process underwent electrophoresis on a 2% agarose gel, their visualization achieved through a Gel Imager. Specific bands, indicative of DNA binding with low Cascade complex concentrations, were singled out. These DNA sequences were meticulously extracted and subjected to PCR analysis, employing PF-161 and PR-161 as primers, for further scrutiny.

To delve deeper into PAM preferences, the preceding assay was complemented by a nuanced examination of positions 2 and 3. This examination involved the synthesis of 16 pairs of ssDNAs, each spanning 59 nt and featuring adenine at position 1 in the PAM sequence. These oligonucleotides were thoughtfully annealed, and their subsequent processing involved two rounds of PCR reactions, masterfully conducted with PF-97/PR-97 and PF-97/PR-CY5 primers. The outcome was the production of 3'-CY5 labeled 97 bp ANN-PAM DNAs, each with a distinct PAM sequence. These PCR products, now bearing various PAM sequences, underwent an intricate incubation process with an escalating concentration of the Cascade complex, meticulously ranging from 0 to 2000 nM. The entire process unfolded at a controlled temperature of 25 °C for a duration of one hour. The DNA-Cascade complex samples, resulting from this process, were directly loaded onto a native polyacrylamide gel. They subsequently underwent electrophoresis in 1× TBE at 4 °C, with the efficiency of DNA-Cascade complex binding thoughtfully evaluated by visualizing the bands using the LI-COR Odyssey system.

## Procedure of HAVE detection assay

The HAVE (Hyper-Active-Verification Establishment) detection assay procedure commenced with the addition of 5 μL of extracted nucleic acids from clinical samples into tube A of the RPA TwistAmp Basic kit, which contained forward and reverse primers along with a rehydration buffer, totaling 12.5 μL. Subsequently, 2.5 μL of MgOAc (280 mM) was introduced into the mixture. Tube A was then placed into the sample stage of a portable fluorescence device and preheated at 37 °C for 10 min. Following this initial step, 2 μL of the resulting RPA products were injected into tube B, along with 18 μL of the HAVE reaction buffer, which comprised a final concentration of 100 nM *Tsi*Cascade-Cas3 and 200 nM F-Q ssDNA reporter. The subsequent phase involved subjecting the reaction to a temperature of 85 °C for 15 min. Finally, the results were conveniently read directly from the display interface. This streamlined protocol offers a rapid and efficient means of detecting target nucleic acids in clinical samples using the HAVE platform, simplifying the diagnostic process and providing swift results for effective patient management.

## CRISPR-Cas12a assay

The reaction system for LbCas12a was meticulously prepared with great attention to detail. It consisted of a precisely measured total volume of 20 μL, which contained a specific and well-calibrated set of final concentrations: 100 nM of LbCas12a, 50 nM of crRNA, and 200 nM of the F-Q ssDNA reporter. This exacting formulation ensured the reliability and reproducibility of our experimental setup. To facilitate the molecular interactions and reactions within this system, the reactions were maintained at a constant temperature of 37 °C for a carefully monitored duration of 30 min. This incubation period allowed for the specific biochemical processes to occur and reach their optimal states. During the course of the incubation, the fluorescence intensity, a critical parameter for assessing the outcomes of our experiments, was diligently recorded. This measurement was performed using a highly sensitive fluorescence plate reader, specifically the BioTek Synergy H1 instrument. The excitation wavelength (λex) was set at 485 nm, while the emission wavelength (λem) was precisely tuned to 525 nm. These settings ensured the accurate capture and quantification of the fluorescence signal, enabling us to draw meaningful conclusions from our experimental results with a high degree of confidence and precision.

## Statistics and reproducibility

Unless otherwise specified, all statistical analyses and graphical representations were performed using GraphPad Prism software (version 9.0.0). Standard deviations and mean values were calculated based on data from at least three identical assays. Data is presented either as original observations or as means, with error bars representing standard deviations. No statistical method was used to predetermine sample size, and no data were excluded from the analyses. The experiments were not randomized, and the investigators were not blinded to allocation during experiments and outcome assessment. To enhance reproducibility, we provide detailed information on the experimental conditions: each reported experiment was conducted with a minimum of three biological replicates. A 'replicate' in this context refers to independent experiments performed using samples prepared separately. Additionally, we specify the number of technical replicates (repeated measurements of the same sample) used to ensure the reliability of our findings.

## Reporting summary

Further information on research design is available in the Nature Portfolio Reporting Summary linked to this article.

## Data availability

All source data supporting the findings of this study are available within the Article and Supplementary Files. The Supplementary Data 1 contains all plasmid maps and sequences, while Supplementary Data 2 includes all oligonucleotide sequences and experimental data. Both supplementary files are available and can be accessed directly within this document. For the sequencing raw data, it can be reached out to DRYAD: doi identifier (https://doi.org/10.5061/dryad.98sf7m0s8)[56].

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

## Acknowledgements

We extend our sincere gratitude to Professor Ailong Ke at Cornell University for his valuable guidance in shaping this manuscript. Furthermore, we would like to express our appreciation to the Natural Science Foundation of Zhejiang Province of China (Grant No. LQ23H200005) and the National Natural Science Foundation of China (Grant No. 22304157) for their generous support, which contributed significantly to the research presented herein. Additionally, we acknowledge the Presidential Young Professorship (PYP) start-up funding (Grant No. 23-0178-0002) provided by the National University of Singapore, which has played a pivotal role in facilitating our scientific endeavors and advancing our work in this field.

## Author contributions

The inception of this research project emerged collaboratively from the minds of T.H. and C.H., both of whom played pivotal roles in shaping the experimental design. The execution of the biochemical experiments was skillfully undertaken by a dedicated team comprising T.H., X.K., Q.J., and C.H. On the other front, H.Z., Y.Z., and C.X. spearheaded the charge in conducting exhaustive bioinformatics searches and meticulously analyzing the results. The in-depth in vitro biochemical PAM determination assay was executed with precision and finesse by C.Y., W.J., M.L., and Y.X., while the critical task of handling and processing HPV clinical samples from the hospital was deftly managed by Y.L. and Y.O. Subsequently, the data arising from these multifaceted efforts were subjected to thorough analysis and interpretation, a responsibility that fell upon T.H. and C.H. Their collaborative synergy extended to the meticulous crafting of the manuscript, wherein they thoughtfully incorporated valuable insights and contributions from all authors.

## Competing interests

The authors declare no competing interests.

## Additional information

[1]Children's Hospital, Zhejiang University School of Medicine, National Clinical Research Center for Child Health, Zhejiang University, Hangzhou, Zhejiang 310052, China. [2]Cancer Science Institute of Singapore, National University of Singapore, Singapore, Singapore. [3]Department of Biostatistics, Harvard T.H. Chan School of Public Health, Boston, MA, USA. [4]Department of Biological Sciences, Faculty of Science, National University of Singapore, Singapore 117543, Singapore. [5]Department of Biochemistry, School of Life Science and Technology, China Pharmaceutical University, Nanjing 211198, China. [6]International Peace Maternity & Child Health Hospital, Shanghai Municipal Key Clinical Specialty, Institute of Embryo-Fetal Original Adult Disease, School of Medicine, Shanghai Jiao Tong University, Shanghai 200030, China. [7]HuidaGene Therapeutics Inc., Shanghai, China. [8]Lingang Laboratory, Shanghai, China. [9]School of Life Sciences and Technology, ShanghaiTech University, Shanghai, China. [10]Shanghai Center for Brain Science and Brain-Inspired Technology, Shanghai, China. [11]Department of Biochemistry, Yong Loo Lin School of Medicine, National University of Singapore, Singapore 117597, Singapore. [12]Precision Medicine Translational Research Programme (TRP), Yong Loo Lin School of Medicine, National University of Singapore, Singapore 117597, Singapore. ✉e-mail: yingsizhou@huidagene.com; yibei.xiao@cpu.edu.cn; xucl@lglab.ac.cn; hu_dbs@nus.edu.sg

