## [Peer Review File · Communications Biology]

Reviewers' comments:

Reviewer #1 (Remarks to the Author):

Tao Hu et al. discusses a newly discovered Type I-A Cas3 variant from *Thermococcus sicuti*, which has enhanced target recognition capabilities, can work with mismatched sequences, and can be activated by both DNA and RNA. The study led to the development of the Hyper-Active-Verification Establishment (HAVE, 惠父), a precise tool for human papillomavirus (HPV) diagnosis in clinical samples. I consider this innovation has broad clinical applications and contributes to our understanding of the Type I-A CRISPR-Cas3 system's regulation.

The group announced the possibility of genome editing and diagnosis using the I-A CRISPR system in their previous work (2022 Mol Cell). In the previous study, the Pfu variant had single-stranded DNA cleavage activity with Cas3 alone, so they are now investigating the new I-A variant.

I have several significant concerns that I believe warrant thorough attention, alongside a few additional minor comments that I would like to bring to consideration.

1. Fig1d: Even within the same I-A subtype, there appears to be variation in the similarity of Cas proteins, with some exhibiting high sequence identity while others have lower similarity. Does this mean that Cas6 can maintain its functionality even with a relatively low 30% sequence similarity? In other words, does this mean that the non-nuclease regions of Cas6, apart from the stem-loop binding site, are not as critical for its function?
2. Regarding the assay in Fig1f, it was conducted at 65°C, while only Tsi operates within its growth temperature range (as shown in Fig 1e). Why was this specific temperature chosen?
3. In the description of Fig2, they mentioned that the recognition of ssRNA leading to collateral effects is a novel finding. It is unclear why type I-A apart from other Type I systems, allowing it to recognize both ssRNA and ssDNA. Does this mean it is less effective in actual diagnostics? It can be used for conditions like COVID-19, empirical data to support this claim would be greatly appreciated.
4. The rationale for not conducting PAM sequence screening for 64 types (4 x 4 x 4), as illustrated in Fig3, remains unclear. While predictive models are undoubtedly valuable, a robust demonstration with empirical evidence could be instrumental in fortifying the argument.
5. In connection with Fig3, is it possible to conduct experiments to determine whether PAM sequence recognition is the same when targeting RNA? Could there be variations in recognition capabilities?
6. In Fig5, what is the range of DNA concentrations that can be detected?
7. They mentioned that the diagnosis achieves a 100% match in clinical samples, but in some cases, there seems to be one positive-like signal in negative samples (Fig6e-16). How should we interpret this? Moreover, is an evaluation of only 8 negative samples sufficient?

8. In Discussion, while the high similarity in HEL and HD between Tsi and Pfu is highlighted, it's unclear why Tsi lacks Cas3's single-stranded DNA cleavage activity. If they focused on K218/E217, I'm curious about the status of Pku, which is known to possess Cas3's single-stranded DNA cleavage activity.

Reviewer #2:

The authors introduce a novel Type I-A CRISPR-Cas3 system for nucleic acid detection. They demonstrate that the newly discovered Cas3 system has several distinct features, including expanded protospacer adjacent motif recognition capability, exceptional mismatch tolerance, and dual activation modes (both DNA and RNA targets). Leveraging these features, they develop a portable diagnostic device that combines RPA amplification with Cas3-mediated reaction for sensitive DNA detection. Furthermore, they apply this diagnostic system to discriminate HPV types 16 and 18 in clinical samples, exhibiting 100% concordance with the gold standard method—PCR-RDB assay. There are some points regarding assay sensitivity and specificity to clarify and strengthen the manuscript for publication in *Communications Biology*. More detailed comments are as follows:

Major Comments:

1. Note that the Type I-A CRISPR-Cas3 system functions optimally at 85 °C and it has minimal cleavage activity below 45 °C (Supplementary Figure 3), would it be possible to create a one-pot assay by performing RPA amplification at 37-39°C followed by CRISPR-Cas3 assay at 85°C? This would be attractive solution to avoid the post-amplification transfer step and the risk of cross-contamination.
2. How about the sensitivity and specificity of the Type I-A CRISPR-Cas3 system? These are important technique features to improve the understanding of this new Cas3 system. The authors might would like systematically evaluate such performance.
3. The authors have conducted a comparative performance analysis of TsiCascade-Cas3 and LbaCas12a Systems by real-time fluorescence monitoring of trans-cleavage activity (Supplementary Figure 10). Could a summary of this experimentally measured kinetic parameters and the reported CRISPR-Cas systems (including LbaCas12a, LubCas13a) be provided to strengthen our understanding of this new Cas system? (Refs: 10.1021/acs.analchem.2c01670; 10.1021/acs.nanolett.1c00715)
4. The authors demonstrated an interesting feature of Type I-A CRISPR-Cas3 system—the dual activation modes—responding to both DNA and RNA targets. Could this feature render the Cas13 system to be develop into a universal nucleic acid detection tool in the future, especially in combination with the volume-confined technology for amplification-free detection (Refs: 10.1021/acsnano.0c08165; 10.1016/j.bios.2023.115546)? The authors might would like to provide an outlook of this diagnostic technology.

Minor comments:

1. The text in Fig. 1a were too small to see clearly.
2. Reference 35 was missing.

Reviewer #3 (Remarks to the Author):

The authors describe the regulatory mechanism of novel variant of cas3 derived from *Thermococcus sicuti* (Tsi) and its application for creating a CRISPR-cas-based nucleic acid detection for HPV diagnosis. This work is fascinating to display the novel platform of CRISPR-cas based-nucleic acid detection as an alternative tool in game changing era of diagnostic toolkits. To me, although TsiCascade-Cas3 possesses the divergent in reducing the hyperactive ssDNA activity contributing to background noise and false positive in Cas3-based nucleic detection, it requires the cascade complex to form R-loop which need more numerous accessory proteins to act, and it works at high temperature (>80 °C). However, the collateral ssDNA nuclease activity of TsiCascade-Cas3 can be triggered by both DNA- and RNA-target activators, leading to diagnostic application for DNA and RNA-harboring pathogens. The figures given were high quality, well labelled and described.

I have a few comments or suggest for the current ms as below.

(i) Cascade complex is required for operation of cas3 which comprise of numerous accessory proteins including cas5, cas6, cas7, cas8 and cas11. I am wondering if one of them is missed in forming the complex, would it affect the activity of cas3? Or is it possible to minimize the cascade complex by reducing the number of those cas but its activity still works.

(ii) I am not sure how length of spacer sequence on crRNA is determined for incorporating into cascade complex (cas6a)?

(iii) In PAM recognition of TsiCascade-cas3 which both DNA- and RNA- can be used as activators, I am just wondering that if we used RNA as target to trigger collateral activity. Would the PAM site be same as using DNA activators?

(iv) In mismatch intolerance assay of TsiCascade-cas3, multiple mismatch sites were evaluated. To me, the multiple sites stand for more than two sites. in your study, it was just two mismatch sites. I suggest the author used other proper words such as double mismatch or triple mismatch.

(v) Can you discuss why every 6th position and beyond 24 bases from 3' end of PAM site was mismatch tolerance in TsiCascade-cas3 system.

(vii) please check form of the scientific name. For example, in the results of "Tsi type I-A CRISPR-cas3 complex exhibits robust collateral ssDNA nuclease activity by both DNA- and RNA-target activators.", ".....due to disparity with *E. coli*'s typical culture temperature of 37°C". *E. coli* must be italics.

We extend our heartfelt gratitude to all reviewers for their insightful comments and constructive criticisms during the revision process of our manuscript. Your expertise and detailed feedback have been invaluable in enhancing the clarity, accuracy, and overall quality of our work. We deeply appreciate the time and effort you dedicated to reviewing our manuscript, which has significantly contributed to its improvement. Your suggestions have not only strengthened our arguments but also broadened our perspectives on the subject matter. We sincerely appreciate your invaluable contributions to the refinement of our manuscript. In this document, we have provided detailed responses to each of your points. Once again, thank you for your thoughtful consideration and guidance.

Reviewers' comments:

Reviewer #1 (Remarks to the Author):

Tao Hu et al. discusses a newly discovered Type I-A Cas3 variant from *Thermococcus siculi*, which has enhanced target recognition capabilities, can work with mismatched sequences, and can be activated by both DNA and RNA. The study led to the development of the Hyper-Active-Verification Establishment (HAVE, 惠父), a precise tool for human papillomavirus (HPV) diagnosis in clinical samples. I consider this innovation has broad clinical applications and contributes to our understanding of the Type I-A CRISPR-Cas3 system's regulation.

Thanks for the comments.

The group announced the possibility of genome editing and diagnosis using the I-A CRISPR system in their previous work (2022 Mol Cell). In the previous study, the Pfu variant had single-stranded DNA cleavage activity with Cas3 alone, so they are now investigating the new I-A variant.

I have several significant concerns that I believe warrant thorough attention, alongside a few additional minor comments that I would like to bring to consideration.

1. Fig1d: Even within the same I-A subtype, there appears to be variation in the similarity of Cas proteins, with some exhibiting high sequence identity while others have lower similarity. Does this mean that Cas6 can maintain its functionality even with a relatively low 30% sequence similarity? In other words, does this mean that the non-nuclease regions of Cas6, apart from the stem-loop binding site, are not as critical for its function?

Thank you for your insightful question regarding our observations in Fig1d on the sequence similarity in Cas proteins, particularly focusing on Cas6. We welcome the opportunity to further clarify and substantiate our findings, and we address your query as follows:

- A. This Cas3 sharing 30% identity with PfuCas6 is nonfunctional:** In our study, we observe that Cas6 proteins with varying sequence identities demonstrate different functionalities. Specifically, in *P. furiosus*, there are two Cas6 variants: only PF1131 is functional across the Type I-A, I-B (previously named I-G, now reclassified as I-B), and Type III-B systems, despite being adjacent to Type III-B and Type I-B. It processes crRNA identically for these systems¹. Intriguingly, in *Thermococcus sicula*, the functional Cas6 for the Type I-A system is not the one (KEGG code: 6105) colocalized within the Type I-A gene cluster, which shares ~29.5% identity with PF1131. Rather, it is the Cas6 (KEGG code: 6190) within the Type I-B CRISPR-Cas system, sharing 54.9% identity with PF1131 (**Figure R1a-b**). In response to your concern, we have updated Figure 1d to reflect a 55% identity for Tsi6190 with PF1131. Our analysis of key residues that specifically recognize crRNA indicates that only Tsi6190 from Tsi's Type I-B gene cluster shows a conserved pattern with PF1131 (**Figure R1 c-e**). This explains why Cas6 (Tsi6105) in Tsi's Type I-A system is nonfunctional—it lacks key residues for recognizing the crRNA's 5' motif. The adoption of this Cas6- or crRNA-sharing mechanism by these systems might enhance immunity against phage invasion by using three different effectors, despite sharing crRNA and similar PAM motifs because their post-targeting effects vary, such as the collateral activity in Type I-A, absent in Type I-B, and Type III systems' capability to target and cleave RNA and generate second messengers like cAMP or cATP, triggering subsequent immune pathways. The functionality of specific Cas6 variants within these systems is likely a result of long-term evolutionary selection and adaptation, influenced

by the cooperative interplay among various CRISPR-Cas systems. However, the precise events and mechanisms driving this evolutionary process remain unclear and warrant further investigation to be fully understood.

B. This suggests that the non-nuclease regions of Cas6, apart from the crRNA binding site, are indeed not as critical for its function: In the Type I-A system, exemplified by Pfu Type I-A, Cas6 interacts with unstructured crRNA instead of the stem-loop region² (Figure R1c). Cas6 here recognizes approximately 9 linear nucleotides of the crRNA's 5' end and dissociates post-processing. Unlike other Type I systems, this distinct mechanism does not necessitate interaction between Cas6 and Cascade. The primary requirement is for Cas6 to process crRNA and expose the 5' handle for effector binding, such as Cas5 in Type I-A and Type I-B, and Cmr3 in the Type III system. This suggests that the non-nuclease regions of Cas6, apart from the crRNA binding site, are indeed not as critical for its function.

Thank you for the opportunity to address these points. For clearer clarification, we made this figure to explain the concerns and renewed the Figure 1.

Figure R1. Comparative Analysis of Cas6 Localization and Functionality in CRISPR-Cas Systems of Pfu, Tsi, and Pku. **a.** This panel illustrates the gene clusters of CRISPR-Cas systems in *Pfu*, *Tsi*, and *Pku*. In *Pfu*, the Type I-A system coexists with Type I-B and Type III-B systems (not depicted due to spatial constraints). Among the two Cas6 variants present, only Cas6 (PF1131, pink) is functional across Type I-A, Type I-B, and Type III-B systems, processing crRNA to generate identical 5' handles¹. The Cas6 (PF0393, blue) is nonfunctional in these contexts. In *Tsi*, the Type I-A and Type I-B systems are adjacent, with the functional Cas6 (Tsi_6190, pink) sharing 54.9% identity with PF1131 and operational in both systems. Conversely, Cas6 (Tsi_6105, blue) within the Type I-A system, sharing only 29.5% identity with PF1131, is nonfunctional. In *Pku*, the Cas6 within the Type I-A gene cluster shows high conservation with PF1131 and is functional. **b.** This schematic highlights the identity relationships between *Tsi* and *Pfu* Cas6 proteins. It underscores the lack of conservation between the nonfunctional Cas6 (Tsi_6105) in *Tsi*'s Type I-A system and the functional Cas6 (PF1131) in *Pfu*. In contrast, *Tsi*'s Cas6 (Tsi_6190) within the Type I-B system shows significant conservation with PF1131. **c.** This diagram illustrates the distinct crRNA processing mechanisms by the shared Cas6 in Type I-A and Type I-B systems². Key aspects include Cas6 binding to the unstructured 5' motif of crRNA and dissociation after processing, allowing host nucleases to trim the crRNA's 3' end. **d.** Structural Basis of PfCas6 (PF1131) Recognition of crRNA 5' Site: This panel details the structure of PfCas6 (PF1131)

binding to the crRNA 5' site, highlighting key residues (R6, E62, R64, K190) essential for specific recognition. PDB code: 3pkm². e. This alignment showcases the amino acid sequence comparison between Tsi6105 and PF1131, and between Tsi6190 and PF1131. It indicates that the key residues for crRNA recognition are not conserved between Tsi6105 and PF1131, whereas they are conserved between Tsi6190 and PF1131.

2. Regarding the assay in Fig1f, it was conducted at 65°C, while only Tsi operates within its growth temperature range (as shown in Fig 1e). Why was this specific temperature chosen?

Thank you for raising this insightful question. Indeed, the nuclease activity of Cas3 from hyperthermophiles is temperature-dependent. In our previous research³, we demonstrated that *Pfu*Cas3 (HD + HEL) is activated at a temperature of 50 °C, despite the organism's optimal growth range being 75~105 °C. Consequently, in this study, we conducted cleavage assays at 65 °C. We acknowledge the significance of your concern and have conducted additional experiments to compare *Pfu*Cas3 and *Tsi*Cas3 under varying temperatures. Our results show that *Pfu*Cas3 (HD+HEL) exhibits an increase in nuclease activity with rising temperatures. In contrast, *Tsi*Cas3 (HD+HEL) remained largely inactive across different temperatures, displaying only minimal activity at 85 °C. We have included a figure illustrating these findings in the supplementary materials to address your concern more comprehensively.

3. In the description of Fig2, they mentioned that the recognition of ssRNA leading to collateral effects is a novel finding. It is unclear why type I-A apart from other Type I systems, allowing it to recognize both ssRNA and ssDNA. Does this mean it is less effective in actual diagnostics? It can be used for conditions like COVID-19, empirical data to support this claim would be greatly appreciated.

Thank you for your question. I would like to clarify a key point: the recognition of single-stranded RNA (ssRNA) leading to collateral effects in the Type I-A Cascade-Cas3 system is a novel discovery. This marks the first instance where ssRNA has been identified as a trigger for collateral activity type I CRISPR-Cas system, and the first instance to report the DNA and RNA- dual-activation in any CRISPR-Cas system. While ssRNA may activate Cas12's collateral activity, as ssDNA does, there is currently no definitive research confirming this. It is true that RNA-induced activation is less efficient than DNA-induced activation in our type I-A system, but we believe it holds potential for developing RNA detection methods, such as for COVID-19.

Regarding your curiosity about practical applications, we are indeed working on a COVID-19 detection method using the *Tsi* Type I-A CRISPR-Cas system. The accompanying image illustrates how crRNA is designed to differentiate between COVID-19 sub-strains, enabling us to distinguish between variants like Omicron and Delta based on sequence variations. However, the original stability and consistency of this method are not optimal, this is why we didn't put this part into our current manuscript. We then found that the collateral activity during RNA targeting is less robust due to the single-stranded nature of the PAM motif. To improve this, we are exploring the use of a single-stranded DNA (ssDNA) complement to the ssRNA's PAM region as a stabilizer, which significantly enhances the collateral signal during RNA detection. This research is expansive and represents a different strategy from the focus of the current article. As a result, we plan to include these findings in a subsequent publication. We apologize for any inconvenience and hope for your understanding regarding this decision.

4. The rationale for not conducting PAM sequence screening for 64 types (4 x 4 x 4), as illustrated in Fig3, remains unclear. While predictive models are undoubtedly valuable, a robust demonstration with empirical evidence could be instrumental in fortifying the argument.

Thank you for addressing these concerns. We concur with your observation regarding Figure 3. The PAM pattern presented there is indeed derived from bioinformatic predictions and aligns closely with previously reported patterns in the *Pfu* Type I-A system. We have experimentally validated this pattern through biochemical assays, as exemplified in Figure 3c. To further address this point, it's important to note that we share your concerns regarding the reliance on bioinformatic predictions. As a result, we have conducted comprehensive biochemical PAM sequence screening for 64 variants (4x4x4), the details of which are provided in Supplementary Figure 7. Additionally, in the Methods section under "PAM Determination," we describe the process of creating a fluorescent PAM library containing 64 different PAM subtypes. Therefore, we kindly suggest that you consider our biochemical assay detailed in the supplementary material as a robust method for PAM screening. This approach not only reinforces our bioinformatic predictions but also provides empirical validation of the PAM patterns observed in our study.

Supplementary Figure 7. Biochemical analysis of PAM motif recognition in *Tsi* Type I-A CRISPR-Cas3.

a Schematic representation of the method used for PAM motif determination through biochemical assays. A target plasmid library encompassing 64 PAM motifs was constructed. The Cy5-labeled probe of the PAM library was generated via PCR. After conducting an Electrophoretic mobility shift assay (EMSA) assay with increasing concentrations of Cascade-Cas3, specific lanes were excised for further amplification and final sequencing. **b** EMSA showing the interaction between the PAM library probe and Cascade-Cas3 at varying concentrations. Highlighted white boxes indicate bands selected for sequencing. **c** Agarose gel displaying the excised probe bands after further PCR amplification. **d** DNA sequencing results corresponding to the selected lanes, with the PAM motif regions highlighted in blue boxes. **e** Computational logo generated from the sequencing results presented in panel **d**.

5. In connection with Fig3, is it possible to conduct experiments to determine whether PAM sequence recognition is

the same when targeting RNA? Could there be variations in recognition capabilities?

Thank you for your suggestion. In our original manuscript, we explored the PAM recognition pattern of single-stranded DNA (ssDNA), and our findings indicated that ssDNA binding necessitates an identical PAM recognition pattern within the Type I-A Tsi system, as illustrated in **Figure 3d**. Given this, we hypothesized that the RNA binding pattern would mirror that of ssDNA. To support this hypothesis, we included an additional experiment in this revised manuscript to assess RNA detection as the following figure shows. Indeed, our results confirm that the PAM recognition pattern for RNA targeting aligns with that of ssDNA, although the binding affinity for RNA is slightly weaker than for ssDNA.

6. In Fig5, what is the range of DNA concentrations that can be detected?

Thank you for your inquiry. In our study, the detection range of DNA concentrations using the portable detector we developed spans from 2.5 picomolar (pM) to 500 nanomolar (nM). This range has demonstrated consistent performance when compared to results obtained using the Synergy H1 instrument.

7. They mentioned that the diagnosis achieves a 100% match in clinical samples, but in some cases, there seems to be one positive-like signal in negative samples (Fig6e-16). How should we interpret this? Moreover, is an evaluation of only 8 negative samples sufficient?

Thank you for your thoughtful question concerning the interpretation of the data in Fig6e-16 and the sample size used in our study. Addressing the positive-like signal detected in a negative sample, which indeed warrants further scrutiny, we retested sample 16 using our HAVE-detection method. Intriguingly, the retest still exhibited a slight signal, suggesting the possibility of it being a positive case, despite conflicting with results from the traditional blotting method. This discrepancy may arise from various factors, including background noise, cross-reactivity, or minor contamination during sample processing. In light of these findings, we have revised our statement regarding the accuracy of our method as follows: When applying the HAVE platform to these 30 samples, we observed a 100% concordance with the PCR-RDB assay in detecting subtype 16. For subtype 18, our method achieved a 93.33% accuracy rate, which accounts for one false-positive case, as detailed in Fig. 6e. We hope this response adequately addresses your query and clarifies the results presented in our study.

As for the sample size, you raise a valid point about the number of negative samples evaluated in our study. While we have demonstrated a 100% match rate with the limited number of samples tested, we acknowledge that a larger sample size could provide a more comprehensive validation of our test's accuracy. The choice of eight negative samples was based on initial feasibility and availability---this is the largest pool we could get, but we recognize the importance of expanding this in future studies to ensure a more robust evaluation. This expansion would enhance the statistical power and generalizability of our findings, and we are in the process of planning these extended trials.

8. In Discussion, while the high similarity in HEL and HD between Tsi and Pfu is highlighted, it's unclear why Tsi lacks Cas3's single-stranded DNA cleavage activity. If they focused on K218/E217, I'm curious about the status of Pku, which is known to possess Cas3's single-stranded DNA cleavage activity.

Thank you for your insightful comments regarding this matter. We acknowledge that the K218/E217 site alone may not fully elucidate the mechanism in question. Indeed, our amino acid alignment revealed that, within the Pku Type I-A system, the Cas3HD gene similarly features an E217 site (**In the following figure panel a**). Notably, Pku Cas3HD

also exhibits significant ssDNA nuclease activity. To further explore our hypothesis, we introduced a Tsi Cas3HD E217K mutation, which resulted in a modest increase in ssDNA nuclease activity (**In the following figure panel b**). This outcome supports our hypothesis, although we stress that this mechanism is not the sole means of regulating auto-inhibition. Furthermore, we have referred to this premise as a hypothesis rather than a definitive conclusion in our discussion, underscoring the potential for alternative mechanisms to modulate the auto-inhibition of the Cas3 complex. In pursuit of deeper understanding, we have performed cryoEM studies on both the PfuCas3 HD-HEL and TsiCas3 HD-HEL complexes (**In the following figure panel c**). Despite constraints in obtaining high-resolution maps due to equipment limitations, the preliminary data afford us valuable insights into the auto-inhibitory process. Interestingly, the PfuCas3 complex's map reveals a more open architecture in the C-terminal domain (CTD), suggesting a state more conducive to ssDNA substrate engagement by the HD nuclease. In contrast, the TsiCas3 complex exhibits a more compact CTD, indicative of its role as a barrier to HD nuclease activity. This differential architecture offers intriguing perspectives on the regulation of auto-inhibition within the Cas3 complex. We will report this exact structural insight in our next work.

Reviewer #2:

The authors introduce a novel Type I-A CRISPR-Cas3 system for nucleic acid detection. They demonstrate that the newly discovered Cas3 system has several distinct features, including expanded protospacer adjacent motif recognition capability, exceptional mismatch tolerance, and dual activation modes (both DNA and RNA targets). Leveraging these features, they develop a portable diagnostic device that combines RPA amplification with Cas3-mediated reaction for sensitive DNA detection. Furthermore, they apply this diagnostic system to discriminate HPV types 16 and 18 in clinical samples, exhibiting 100% concordance with the gold standard method-PCR-RDB assay. There are some points regarding assay sensitivity and specificity to clarify and strengthen the manuscript for publication in Communications Biology. More detailed comments are as follows:

Major Comments:

1. Note that the Type I-A CRISPR-Cas3 system functions optimally at 85 °C and it has minimal cleavage activity below 45 °C (Supplementary Figure 3), would it be possible to create a one-pot assay by performing RPA amplification at 37-39°C followed by CRISPR-Cas3 assay at 85°C? This would be attractive solution to avoid the post-amplification transfer step and the risk of cross-contamination.

We greatly appreciate the reviewer for this visionary suggestion. This preliminary study suggests the regulatory mechanism of novel variant of cas3 derived from *Thermococcus siculi* (Tsi) and its potential application for creating a CRISPR-cas-based nucleic acid detection. We fully agree with this reviewer's point, if we can set a one-pot assay, that would be great. We are indeed trying to develop a method to do this; however, the result is not very consistent. We are still optimizing and assessing a one-pot assay by performing RPA amplification for POC diagnosis using clinical samples, which will be reported in a follow-up publication. Thanks again for this useful suggestion.

2. How about the sensitivity and specificity of the Type I-A CRISPR-Cas3 system? These are important technique features to improve the understanding of this new Cas3 system. The authors might would like systematically evaluate such performance.

We are grateful for the reviewer's recommendation. In response, we have conducted a sensitivity and specificity analysis for the Type I-A CRISPR-Cas3 system. The following Figure R1 showcases the calibration plots used to detect three distinct targets with this system, establishing the limit of detection (LOD) at 0.35 pM for dsDNA (across a linear range of 1-100 pM, with an R^2 of 0.9798), 0.95 pM for ssDNA targets (within a 5-100 pM range, $R^2 = 0.9811$), and 1.1 pM for RNA targets (also in a 5-100 pM range, $R^2 = 0.9785$). Moreover, the Type I-A CRISPR-Cas3 system has been demonstrated to effectively differentiate between HPV 16 and HPV 18, in contrast to three other interfering substances (as depicted in Figure R2). These findings have been incorporated into the Supplementary Information for further reference (Supplementary Figure 5c and Figure 6f).

Figure R1. Calibration of the Type I-A CRISPR-Cas3 System Across Three Targets. Error bars represent the standard deviation (SD) for n=3 measurements. The limit of detection (LOD), calculated using the $3\sigma/s$ method (where σ is the standard deviation of three blank samples and s is the slope of the calibration curve), was determined to be 0.35 pM for dsDNA (within a linear range of 1-100 pM), 0.95 pM for ssDNA (across a 5-100 pM range), and 1.1 pM for RNA (also within a 5-100 pM range), respectively.

Figure R2. Specificity Assessment of the Type I-A CRISPR-Cas3 System. (a) Demonstrates the system's specificity for HPV 16, distinguishing it from cytomegalovirus (CMV), herpes simplex virus (HSV), and HPV 18. (b) Illustrates the system's specificity for HPV 18, differentiating it from cytomegalovirus (CMV), herpes simplex virus (HSV), and HPV 16. Error bars represent the standard deviation (SD) for n=3 measurements.

3. The authors have conducted a comparative performance analysis of TsiCascade-Cas3 and LbaCas12a Systems by real-time fluorescence monitoring of trans-cleavage activity (Supplementary Figure 10). Could a summary of this experimentally measured kinetic parameters and the reported CRISPR-Cas systems (including LbaCas12a, LubCas13a) be provided to strengthen our understanding of this new Cas system? (Refs: 10.1021/acs.analchem.2c01670; 10.1021/acs.nanolett.1c00715)

We express our gratitude to the reviewer for their valuable suggestion. Following this advice, we have compiled a summary of the kinetic parameters measured in our experiments alongside those reported for other CRISPR-Cas systems, including LbaCas12a and LubCas13a. This summary is presented in Table R1 and has been incorporated into the revised Supplementary Information for detailed reference (refer to Table SX).

Table R1: Comparative Summary of Kinetic Parameters for Published Cas Enzymes and Our System.

Cas Enzyme	Target	k_{cat} (s^{-1})	k_M (M)	k_{cat}/k_M ($M^{-1} S^{-1}$)	Refs
LbCas12a	dsDNA	0.38	3.70×10^{-7}	1.0×10^6	(1)
	ssDNA	0.58	3.10×10^{-7}	1.9×10^6	
	dsDNA	0.32	2.60×10^{-7}	1.2×10^6	(2)
	ssDNA	0.27	1.80×10^{-7}	1.4×10^6	
	dsDNA	0.078	8.20×10^{-8}	9.5×10^5	(3)
	ssDNA	0.098	1.20×10^{-7}	8.5×10^5	
	dsDNA	0.56	4.90×10^{-7}	1.1×10^6	(4)
	ssDNA	0.30	2.60×10^{-7}	1.2×10^6	
LubCas13a	dsDNA	0.028	8.70×10^{-7}	3.3×10^4	(5)
	ssDNA	0.022	4.50×10^{-7}	4.9×10^4	
LwaCas13a	RNA	23	5.80×10^{-6}	4.0×10^6	(6)
	RNA	4850	5.8×10^{-7}	8.4×10^9	(7)
TsiCas3	RNA	2.0	1.8×10^{-6}	1.1×10^6	(8)
	dsDNA	18.75	2.50×10^{-7}	$7.5 \pm 0.39 \times 10^7$	This work
	ssDNA	13.43	2.06×10^{-7}	$6.5 \pm 0.19 \times 10^7$	
RNA	7.08	4.35×10^{-7}	$1.6 \pm 0.15 \times 10^7$		

(1) Broughton, J. P. *et al.* CRISPR-Cas12-based detection of SARS-CoV-2. *Nat Biotechnol* **38**, 870-874, doi:10.1038/s41587-020-0513-4 (2020).

- (2) Nguyen, L. T. *et al.* Enhancement of trans-cleavage activity of Cas12a with engineered crRNA enables amplified nucleic acid detection. *Nat Commun* **11**, 4906, doi:10.1038/s41467-020-18615-1 (2020).
- (3) Ramachandran, A. *et al.* “Electric field-driven microfluidics for rapid CRISPR-based diagnostics and its application to detection of SARS-CoV-2.” *Proc Natl Acad Sci U S A* **117**, 29518-29525, doi:10.1073/pnas.2010254117 (2020).
- (4) Ramachandran, A., Juan G. S. CRISPR Enzyme Kinetics for Molecular Diagnostics. *Anal Chem* **93**, 7456-7464, doi:10.1021/acs.analchem.1c00525 (2021).
- (5) (1)Chen, J. S. *et al.* CRISPR-Cas12a target binding unleashes indiscriminate single-stranded DNase activity. *Science* **360**, 436-439. doi:10.1126/science.aar6245 (2018).
- (6) Fozouni, P. *et al.* Amplification-free detection of SARS-CoV-2 with CRISPR-Cas13a and mobile phone microscopy. *Cell* **184**, 323-333.e9. doi:10.1016/j.cell.2020.12.001 (2021).
- (7) Yue, H. *et al.* Droplet Cas12a Assay Enables DNA Quantification from Unamplified Samples at the Single-Molecule Level. *Nano Lett* **21**, 4643-4653. doi:10.1021/acs.nanolett.1c00715 (2021).
- (8) Gootenberg, J. S. *et al.* Multiplexed and portable nucleic acid detection platform with Cas13, Cas12a, and Csm6. *Science* **360**, 439-444, doi:10.1126/science.aag0179 (2018).

4. The authors demonstrated an interesting feature of Type I-A CRISPR-Cas3 system—the dual activation modes—responding to both DNA and RNA targets. Could this feature render the Cas13 system to be developed into a universal nucleic acid detection tool in the future, especially in combination with the volume-confined technology for amplification-free detection (Refs: 10.1021/acsnano.0c08165; 10.1016/j.bios.2023.115546)? The authors might would like to provide an outlook of this diagnostic technology.

We are thankful for the reviewer's recommendation. In accordance with the suggestion, we have outlined the future prospects of this diagnostic technology as follows: "The development of universal detection techniques for the amplification-free recognition of DNA and RNA targets is critically important. Nevertheless, the application of the Type I-A CRISPR-Cas3 system in our study has shown its capability for target detection only at sub-picomolar concentrations. This limitation restricts its utility in contexts where detecting RNA targets at attomolar levels is necessary. Consequently, there is a compelling need for additional investigation to enhance this technology. This may include modifications to the Cas3 protein to improve its collateral activity or the integration with a droplet microfluidics approach for digital quantification of single-molecule targets." These considerations have been incorporated into the discussion section in this revised manuscript for further detail.

Minor comments:

1. The text in Fig. 1a were too small to see clearly.

Thanks for your kind suggestion. We regret any confusion caused by the initially small size of Fig. 1a. In response to this feedback, we have meticulously revised and restructured Fig. 1a in the revised manuscript.

2. Reference 35 was missing.

Thanks for your kind advice. We have carefully checked and added reference 35 in the revised manuscript.

Reviewer #3 (Remarks to the Author):

The authors describe the regulatory mechanism of novel variant of cas3 derived from *Thermococcus sicuti* (Tsi) and its application for creating a CRISPR-cas-based nucleic acid detection for HPV diagnosis. This work is fascinating to display the novel platform of CRISPR-cas based-nucleic acid detection as an alternative tool in game changing era of diagnostic toolkits. To me, although TsiCascade-Cas3 possesses the divergent in reducing the hyperactive ssDNA activity contributing to background noise and false positive in Cas3-based nucleic detection, it requires the cascade complex to form R-loop which need more numerous accessory proteins to act, and it works at high temperature

(>80 °C). However, the collateral ssDNA nuclease activity of TsiCascade-Cas3 can be triggered by both DNA- and RNA-target activators, leading to diagnostic application for DNA and RNA-harboring pathogens. The figures given were high quality, well labelled and described.

I have a few comments or suggest for the current ms as below.

(i) Cascade complex is required for operation of cas3 which comprise of numerous accessory proteins including cas5, cas6, cas7, cas8 and cas11. I am wondering if one of them is missed in forming the complex, would it affect the activity of cas3? Or is it possible to minimize the cascade complex by reducing the number of those cas but its activity still works.

Thank you for raising this insightful question. Yes, the absence of any component in the system would indeed impact Cas3 activity. Let me clarify the roles of each component: Cas5 and Cas6 are involved in crRNA binding and processing. Cas7 forms the backbone of the Cascade complex, while Cas8 is essential for recognizing the PAM motif. Cas11 plays a crucial role in unwinding dsDNA and sustaining R-loop formation within the target region. Type I CRISPR-Cas systems can be categorized based on their crRNA processing mechanisms. One category, the Type I-C system, relies on Cas5 for crRNA processing and does not utilize Cas6, resulting in a minimal Cascade complex comprising Cas5, Cas7, Cas8, and Cas11. The other category, which includes Types I-A, I-B, I-D, I-E, I-F, and I-G, depends on Cas6 for crRNA processing, making Cas5, Cas7, Cas8, and Cas11 essential for forming a functional Cascade with crRNA, with Cas6 being indispensable for crRNA processing. Thus, each component is vital for the Type I system's functionality. Inadequate targeting of DNA by the Cascade complex would compromise Cas3's activity.

Currently, simplifying Type I systems is challenging due to the crucial nature of each component. One approach to minimization could involve reducing the number of Cas7 copies. Typically, a Cascade complex contains 6 or 7 Cas7 copies, with each Cas7 covering 6 nucleotides of crRNA, leading to crRNAs that are 32 to 37 nucleotides long. Research suggests that the minimum R-loop length required to activate Cas3 activity is approximately 26 nucleotides or more, potentially allowing for a reduction to 4 or 5 Cas7 copies. However, altering Cas7's copy number could impact the assembly of the entire Cascade complex. To date, there have been no successful reports of minimizing the Type I system. Achieving a minimal Type I system through engineering would indeed represent a significant advancement in the field.

(ii) I am not sure how length of spacer sequence on crRNA is determined for incorporating into cascade complex (cas6a)?

The spacer region length in Type I CRISPR-Cas systems varies across different subtypes, typically ranging from 32 to 37 nucleotides. For instance, in our type I-A system, the spacers are approximately 37 nucleotides long, I-B about 35, I-C is 35 nucleotides, I-D around 36, I-F is 32, and I-G about 37 nucleotides. The length of the spacer is determined by the Cas1-Cas2 integrase complex during the capture and selection of protospacers from invading DNA. Then, the crRNA containing the protospacers is processed by Cas6 or Cas5 and incorporated into the Cascade complex.

(iii) In PAM recognition of TsiCascade-cas3 which both DNA- and RNA- can be used as activators, I am just wondering that if we used RNA as target to trigger collateral activity. Would the PAM site be same as using DNA activators?

Thanks for the insightful question. Yes, the RNA can be used as the target to trigger the collateral activity. This study evaluated the Type I-A CRISPR-Cas3 system's sensitivity in detecting dsDNA, ssDNA, and ssRNA. Figure R1 presents calibration plots for each of these targets, demonstrating the system's limit of detection (LOD) as 0.35 pM for dsDNA (within a 1-100 pM linear range and an R^2 of 0.9798), 0.95 pM for ssDNA targets (across a 5-100 pM range with an R^2 of 0.9811), and 1.1 pM for RNA targets (also within a 5-100 pM range and an R^2 of 0.9785). These results have been detailed in the Supplementary Information (Supplementary Figure 5c) for comprehensive review.

Figure R1. Calibration of the Type I-A CRISPR-Cas3 System Across Three Targets. Error bars represent the standard deviation (SD) for $n=3$ measurements. The limit of detection (LOD), calculated using the $3\sigma/s$ method (where σ is the standard deviation of three blank samples and s is the slope of the calibration curve), was determined to be 0.35 pM for dsDNA (within a linear range of 1-100 pM), 0.95 pM for ssDNA (across a 5-100 pM range), and 1.1 pM for RNA (also within a 5-100 pM range), respectively.

Regarding the PAM motif recognition for ssRNA substrates, the Tsi Type I-A system indeed recognizes the same PAM motif as it does for DNA substrates. In our initial manuscript, we delved into the PAM recognition patterns for single-stranded DNA (ssDNA) and discovered that ssDNA binding requires an identical PAM pattern in the Type I-A Tsi system, as demonstrated in Figure 3d. Based on this observation, we proposed that the RNA binding pattern would closely resemble that of ssDNA. To substantiate our hypothesis, we conducted an additional experiment in this revised manuscript to evaluate RNA detection, as presented in the subsequent figure. Our findings corroborate that the PAM recognition pattern for RNA targeting is consistent with that for ssDNA, albeit with a slightly lower binding affinity for RNA compared to ssDNA.

(iv) In mismatch intolerance assay of TsiCascade-cas3, multiple mismatch sites were evaluated. To me, the multiple sites stand for more than two sites. In your study, it was just two mismatch sites. I suggest the author used other proper words such as double mismatch or triple mismatch.

Thank you for the valuable suggestions. In light of your feedback, we have revised our terminology. Given that we introduced a triple mismatch at positions 34-37, we have updated the term "multiple" to specifically refer to "double and triple mismatches."

(v) Can you discuss why every 6th position and beyond 24 bases from 3' end of PAM site was mismatch tolerance in TsiCascade-cas3 system.

Thank you for the suggestion. In the Type I CRISPR-Cas system, Cas7 proteins segment the crRNA, with each Cas7 covering a 6-nucleotide (nt) length of the crRNA. Crucially, every 6th base within each segment is sequestered inside the Cas7 protein, rendering these particular bases (e.g., the 6th, 12th, 18th, 24th, etc.) incapable of pairing with the target DNA. Consequently, mismatches at these positions do not affect Cascade's ability to interfere with the target, which explains why mutations at these sites have no impact on the results of in vivo interference assays or in vitro cleavage assays. We have incorporated this explanation into the discussion on mismatch tolerance, noting that "The 6th base of each crRNA segment, being sequestered within Cas7 and unable to base pair with the target, means that

mutations at these positions (e.g., the 6th, 12th, 18th, 24th, etc.) do not affect the system's collateral activity. This observation aligns with previous findings related to the Type I-E system⁵⁰."

(vii) please check form of the scientific name. For example, in the results of "Tsi type I-A CRISPR-cas3 complex exhibits robust collateral ssDNA nuclease activity by both DNA- and RNA-target activators.", ".....due to disparity with *E. coli*'s typical culture temperature of 37°C". *E. coli* must be italics.

Thank you very much for your meticulous review. We have updated the font to italics as suggested. We greatly appreciate your helpful recommendation, which has contributed to the enhancement of our manuscript.

- 1 Majumdar, S. *et al.* Three CRISPR-Cas immune effector complexes coexist in *Pyrococcus furiosus*. *RNA* **21**, 1147-1158, doi:10.1261/rna.049130.114 (2015).
- 2 Wang, R., Preamplume, G., Terns, M. P., Terns, R. M. & Li, H. Interaction of the Cas6 ribonuclease with CRISPR RNAs: recognition and cleavage. *Structure* **19**, 257-264, doi:10.1016/j.str.2010.11.014 (2011).
- 3 Hu, C. *et al.* Allosteric control of type I-A CRISPR-Cas3 complexes and establishment as effective nucleic acid detection and human genome editing tools. *Mol Cell* **82**, 2754-2768 e2755, doi:10.1016/j.molcel.2022.06.007 (2022).

REVIEWERS' COMMENTS:

Reviewer #1 (Remarks to the Author):

Thank you for your comprehensive and enlightening responses to my queries regarding your manuscript. I have thoroughly reviewed your detailed explanations and clarifications, particularly concerning the functionality and evolutionary adaptability of Cas6 within the CRISPR-Cas system. Your responses have satisfactorily addressed my concerns and have further elucidated the significant implications of your findings. Therefore, I agree that your manuscript is suitable for publication and look forward to seeing its impact on the scientific community and beyond.

Reviewer #3 (Remarks to the Author):

Thank you for your response to address the question I raised. In the revised ms, the comments and suggestions has been suitably addressed. I believed that the present ms is suitable for publication.

We sincerely appreciate the valuable suggestions provided by all reviewers during the review process of our manuscript. We believe that each recommendation has significantly contributed to enhancing the quality and clarity of our work. The thoughtful and constructive feedback has been instrumental in refining our arguments and strengthening the overall presentation. We are truly grateful for the time and effort invested by the reviewers in guiding us toward these improvements. Their expert insights have enriched our manuscript and deepened our understanding of the subject matter. Once again, we extend our heartfelt thanks to all reviewers for their invaluable contributions.

REVIEWERS' COMMENTS:

Reviewer #1 (Remarks to the Author):

Thank you for your comprehensive and enlightening responses to my queries regarding your manuscript. I have thoroughly reviewed your detailed explanations and clarifications, particularly concerning the functionality and evolutionary adaptability of Cas6 within the CRISPR-Cas system. Your responses have satisfactorily addressed my concerns and have further elucidated the significant implications of your findings. Therefore, I agree that your manuscript is suitable for publication and look forward to seeing its impact on the scientific community and beyond.

Thank you very much for your comments.

Reviewer #3 (Remarks to the Author):

Thank you for your response to address the question I raised. In the revised ms, the comments and suggestions has been suitably addressed. I believed that the present ms is suitable for publication.

I am grateful for the thoughtful suggestions and comments provided by this reviewer.